# Robust No-Regret Learning in Min-Max Stackelberg Games

**Denizalp Goktas[1], Jiayi Zhao [2], Amy Greenwald[1]**

[1] Brown University, Computer Science Department
{denizalp_goktas, amy_greenwald}@brown.edu
[2] Pomona Colege, Computer Science Department
jzae2019@mymail.pomona.edu

## Abstract

The behavior of no-regret learning algorithms is well understood in two-player min-max (i.e, zero-sum) games. In this paper, we investigate the behavior of no-regret learning in min-max games *with dependent strategy sets*, where the strategy of the first player constrains the behavior of the second. Such games are best understood as sequential, i.e., min-max Stackelberg, games. We consider two settings, one in which only the first player chooses their actions using a no-regret algorithm while the second player best responds, and one in which both players use no-regret algorithms. For the former case, we show that no-regret dynamics converge to a Stackelberg equilibrium. For the latter case, we introduce a new type of regret, which we call Lagrangian regret, and show that if both players minimize their Lagrangian regrets, then play converges to a Stackelberg equilibrium. We then observe that online mirror descent (OMD) dynamics in these two settings correspond respectively to a known nested (i.e., sequential) gradient descent-ascent (GDA) algorithm and a new simultaneous GDA-like algorithm, thereby establishing convergence of these algorithms to Stackelberg equilibrium. Finally, we analyze the robustness of OMD dynamics to perturbations by investigating *dynamic* min-max Stackelberg games. We prove that OMD dynamics are robust for a large class of dynamic min-max games with independent strategy sets. In the dependent case, we demonstrate the robustness of OMD dynamics experimentally by simulating them in dynamic Fisher markets, a canonical example of a min-max Stackelberg game with dependent strategy sets.

## 1 Introduction

Min-max optimization problems (i.e., zero-sum games) have been attracting a great deal of attention recently because of their applicability to problems in fairness in machine learning (Dai et al. 2019; Edwards and Storkey 2016; Madras et al. 2018; Sattigeri et al. 2018), generative adversarial imitation learning (Cai et al. 2019; Hamedani et al. 2018), reinforcement learning (Dai et al. 2018), generative adversarial learning (Sanjabi et al. 2018a), adversarial learning (Sinha et al. 2020), and statistical learning, e.g., learning parameters of exponential families (Dai et al. 2019). These problems are often modelled as **min-max games**, i.e., constrained min-max optimization prob-

lems of the form: $\min_{\boldsymbol{x} \in X} \max_{\boldsymbol{y} \in Y} f(\boldsymbol{x}, \boldsymbol{y})$, where $f : X \times Y \rightarrow \mathbb{R}$ is continuous, and $X \subset \mathbb{R}^n$ and $Y \subset \mathbb{R}^m$ are non-empty and compact. In **convex-concave min-max games**, where $f$ is convex in $\boldsymbol{x}$ and concave in $\boldsymbol{y}$, von Neumann and Morgenstern's seminal minimax theorem holds (Neumann 1928): i.e., $\min_{\boldsymbol{x} \in X} \max_{\boldsymbol{y} \in Y} f(\boldsymbol{x}, \boldsymbol{y}) = \max_{\boldsymbol{y} \in Y} \min_{\boldsymbol{x} \in X} f(\boldsymbol{x}, \boldsymbol{y})$, guaranteeing the existence of a saddle point, i.e., a point that is simultaneously a minimum of $f$ in the $\boldsymbol{x}$-direction and a maximum of $f$ in the $\boldsymbol{y}$-direction. This theorem allows us to interpret the optimization problem as a simultaneous-move, zero-sum game, where $\boldsymbol{y}^*$ (resp. $\boldsymbol{x}^*$) is a best-response of the outer (resp. inner) player to the other's action $\boldsymbol{x}^*$ (resp. $\boldsymbol{y}^*$), in which case a saddle point is also called a minimax point or a Nash equilibrium.

In this paper, we study **min-max Stackelberg games** (Goktas and Greenwald 2021), i.e., constrained min-max optimization problems *with dependent feasible sets* of the form: $\min_{\boldsymbol{x} \in X} \max_{\boldsymbol{y} \in Y : \boldsymbol{g}(\boldsymbol{x}, \boldsymbol{y}) \geq \boldsymbol{0}} f(\boldsymbol{x}, \boldsymbol{y})$, where $f : X \times Y \rightarrow \mathbb{R}$ is continuous, $X \subset \mathbb{R}^n$ and $Y \subset \mathbb{R}^m$ are non-empty and compact, and $\boldsymbol{g}(\boldsymbol{x}, \boldsymbol{y}) = (g_1(\boldsymbol{x}, \boldsymbol{y}), \ldots, g_K(\boldsymbol{x}, \boldsymbol{y}))^T$ with $g_k : X \times Y \rightarrow \mathbb{R}$. Goktas and Greenwald observe that the minimax theorem does not hold in these games (2021). As a result, such games are more appropriately viewed as sequential, i.e., Stackelberg, games for which the relevant solution concept is the Stackelberg equilibrium,[1] where the outer player chooses $\hat{\boldsymbol{x}} \in X$ before the inner player responds with their choice of $\boldsymbol{y}(\hat{\boldsymbol{x}}) \in Y$ s.t. $\boldsymbol{g}(\hat{\boldsymbol{x}}, \boldsymbol{y}(\hat{\boldsymbol{x}})) \geq \boldsymbol{0}$. In these games, the outer player seeks to minimize their loss, assuming the inner player chooses a feasible best response: i.e., the outer player's objective, also known as their **value function** in the economics literature (Milgrom and Segal 2002), is defined as $V_X(\boldsymbol{x}) = \max_{\boldsymbol{y} \in Y : \boldsymbol{g}(\boldsymbol{x}, \boldsymbol{y}) \geq \boldsymbol{0}} f(\boldsymbol{x}, \boldsymbol{y})$. The inner player's value function, $V_Y : X \rightarrow \mathbb{R}$, which they seek to maximize, is simply the objective function given the outer player's action $\hat{\boldsymbol{x}}$: i.e., $V_Y(\boldsymbol{y}; \hat{\boldsymbol{x}}) = f(\hat{\boldsymbol{x}}, \boldsymbol{y})$.

Goktas and Greenwald (2021) proposed a polynomial-time first-order method by which to compute Stackelberg

---

[1]One could also view such games as pseudo-games (also known as abstract economies) (Arrow and Debreu 1954), in which players move simultaneously under the unreasonable assumption that the moves they make will satisfy the game's dependency constraints. Under this view, the relevant solution concept is generalized Nash equilibrium (Facchinei and Kanzow 2007, 2010).

equilibria, which they called **nested gradient descent ascent (GDA)**. This method can be understood as an algorithm a third party might run to find an equilibrium, or as a game dynamic that the players might employ if their long-run goal were to reach an equilibrium. Rather than assume that players are jointly working towards the goal of reaching an equilibrium, it is often more reasonable to assume that they play so as to not regret their decisions: i.e., that they employ a **no-regret learning algorithm**, which minimizes their loss in hindsight. It is well known that when both players in a min-max game are no-regret learners, the players' strategy profile over time converges to a Nash equilibrium in average iterates: i.e., empirical play converges to a Nash equilibrium (e.g., (Freund and Schapire 1996)).

In this paper, we investigate no-regret learning dynamics in min-max Stackelberg games. We assume both pessimistic and optimistic settings. In the pessimistic setting, the outer player is a no-regret learner while the inner player best responds; in the optimistic setting, both players are no-regret learners. In the pessimistic case, we show that if the outer player uses a no-regret algorithm that achieves $\varepsilon$-pessimistic regret after $T$ iterations, then the outer player's empirical play converges to their $\varepsilon$-Stackelberg equilibrium strategy. In the optimistic case, we introduce a new type of regret, which we call Lagrangian regret, which assumes access to a solution oracle for the optimal KKT multipliers of the game's constraints. We then show that if both players use no-regret algorithms that achieve $\varepsilon$-Lagrangian regret after $T$ iterations, the players' empirical play converges to an $\varepsilon$-Stackelberg equilibrium.

We then restrict our attention to online mirror descent (OMD) dynamics, which yield two algorithms, namely max-oracle gradient descent (Jin, Netrapalli, and Jordan 2020) and nested GDA (Goktas and Greenwald 2021) in the pessimistic setting, and a new simultaneous GDA-like algorithm (Nedic and Ozdaglar 2009) in the optimistic setting, which we call Lagrangian GDA (LGDA). Convergence of the former two algorithms in $O(1/\varepsilon^2)$ iterations then follows from our previous theorems. Additionally, the iteration complexity of $O(1/\varepsilon^2)$ suggests the superiority of LGDA over nested-GDA when a Lagrangian solution oracle exists, since nested-GDA converges in $O(1/\varepsilon^3)$ iterations (Goktas and Greenwald 2021), while LGDA converges in $O(1/\varepsilon^2)$ iterations, assuming the objective function is only Lipschitz continuous.

Finally, we analyze the robustness of OMD dynamics to perturbations by investigating *dynamic* min-max Stackelberg games. We prove that OMD dynamics are robust, in that even when the game changes with each iteration of the algorithm, OMD dynamics track the changing equilibrium closely for a large class of dynamic min-max games with independent strategy sets. In the dependent strategy set case, we demonstrate the robustness of OMD dynamics experimentally by simulating them in dynamic Fisher markets, a canonical example of a min-max Stackelberg game (with dependent strategy sets). Even when the Fisher market changes with each iteration, our OMD dynamics are able to track the changing equilibria closely. Our findings can be summarized as follows:

- In min-max Stackelberg games, when the outer player is a no-regret learner and the inner-player best-responds, the average of the outer player's strategies converges to their Stackelberg equilibrium strategy.
- We introduce a new type of regret we call Lagrangian regret and show that in min-max Stackelberg games when both players minimize Lagrangian regret, the average of the players' strategies converges to a Stackelberg equilibrium.
- We provide novel convergence guarantees for two known algorithms, max-oracle gradient descent and nested gradient descent ascent, to an $\varepsilon$-Stackelberg equilibrium in $O(1/\varepsilon^2)$ in average iterates.
- We introduce a new simultaneous GDA-like algorithm and prove that its average iterates converge to an $\varepsilon$-Stackelberg equilibrium in $O(1/\varepsilon^2)$ iterations.
- We prove that max-oracle gradient descent and simultaneous GDA are robust to perturbations in a large class of min-max games (with independent strategy sets).
- We run experiments with Fisher markets which suggest that max-oracle gradient descent and simultaneous GDA are robust to perturbations in these min-max Stackelberg games (with dependent strategy sets).

We provide a review of related work in Appendix BThis paper is organized as follows. In the next section, we present the requisite mathematical preliminaries. In Section 3, we present no-regret learning dynamics that converge in a large class of min-max Stackelberg games. In Section 4, we study the convergence and robustness properties of a particular no-regret learning algorithm, namely online mirror descent, in min-max Stackelberg games.

## 2  Mathematical Preliminaries

Our notational conventions can be found in Appendix A.

**Game Definitions** A **min-max Stackelberg game**, $(X, Y, f, \boldsymbol{g})$, is a two-player, zero-sum game, where one player, who we call the outer, or $\boldsymbol{x}$-, player (resp. the inner, or $\boldsymbol{y}$-, player), is trying to minimize their loss (resp. maximize their gain), defined by a continuous **objective function** $f : X \times Y \to \mathbb{R}$, by taking an action from their **strategy set** $X \subset \mathbb{R}^n$, and (resp. $Y \subset \mathbb{R}^m$) s.t. $\boldsymbol{g}(\boldsymbol{x}, \boldsymbol{y}) \geq 0$ where $\boldsymbol{g}(\boldsymbol{x}, \boldsymbol{y}) = (g_1(\boldsymbol{x}, \boldsymbol{y}), \ldots, g_K(\boldsymbol{x}, \boldsymbol{y}))^T$ with $g_k : X \times Y \to \mathbb{R}$ continuous. A strategy profile $(\boldsymbol{x}, \boldsymbol{y}) \in X \times Y$ is said to be **feasible** iff for all $k \in [K]$, $g_k(\boldsymbol{x}, \boldsymbol{y}) \geq 0$. The function $f$ maps a pair of actions taken by the players $(\boldsymbol{x}, \boldsymbol{y}) \in X \times Y$ to a real value (i.e., a payoff), which represents the loss (resp. the gain) of the $\boldsymbol{x}$-player (resp. the $\boldsymbol{y}$-player). A min-max game is said to be convex-concave if the objective function $f$ is convex-concave.

One way to see this game is as a **Stackelberg game**, i.e., a sequential game with two players, where WLOG, we assume that the minimizing player moves first and the maximizing player moves second. The relevant solution concept for Stackelberg games is the **Stackelberg equilibrium**: A strategy profile $(\boldsymbol{x}^*, \boldsymbol{y}^*) \in X \times Y$ s.t. $\boldsymbol{g}(\boldsymbol{x}^*, \boldsymbol{y}^*) \geq \mathbf{0}$ is an $(\epsilon, \delta)$-Stackelberg equilibrium if

$\max_{\boldsymbol{y} \in Y: \boldsymbol{g}(\boldsymbol{x}^*, \boldsymbol{y}) \geq 0} f(\boldsymbol{x}^*, \boldsymbol{y}) - \delta \leq f(\boldsymbol{x}^*, \boldsymbol{y}^*)$
$\leq \min_{\boldsymbol{x} \in X} \max_{\boldsymbol{y} \in Y: \boldsymbol{g}(\boldsymbol{x}, \boldsymbol{y}) \geq 0} f(\boldsymbol{x}, \boldsymbol{y}) + \epsilon$. Intuitively, a $(\varepsilon, \delta)$-Stackelberg equilibrium is a point at which the $\boldsymbol{x}$-player's (resp. $\boldsymbol{y}$-player's) payoff is no more than $\varepsilon$ (resp. $\delta$) away from its optimum. A $(0, 0)$-Stackelberg equilibrium is guaranteed to exist in min-max Stackelberg games (Goktas and Greenwald 2021). Note that when $\boldsymbol{g}(\boldsymbol{x}, \boldsymbol{y}) \geq \boldsymbol{0}$, for all $\boldsymbol{x} \in X$ and $\boldsymbol{y} \in Y$, the game reduces to a min-max game (with independent strategy sets), for which, by the min-max theorem, a Nash equilibrium is guaranteed to exist (Neumann 1928).

In a min-max Stackelberg game, the outer player's **best-response set** $\mathrm{BR}_X \subset X$, defined as $\mathrm{BR}_X = \arg\min_{\boldsymbol{x} \in X} V_X(\boldsymbol{x})$, is independent of the inner player's strategy, while the inner player's **best-response correspondence** $\mathrm{BR}_Y : X \rightrightarrows Y$, defined as $\mathrm{BR}_Y(\boldsymbol{x}) = \arg\max_{\boldsymbol{y} \in Y: \boldsymbol{g}(\boldsymbol{x}, \boldsymbol{y}) \geq 0} V_Y(\boldsymbol{y}; \boldsymbol{x})$, depends on the outer player's strategy. A $(0, 0)$-Stackelberg equilibrium $(\boldsymbol{x}^*, \boldsymbol{y}^*) \in X \times Y$ is then a tuple of strategies such that $(\boldsymbol{x}^*, \boldsymbol{y}^*) \in \mathrm{BR}_X \times \mathrm{BR}_Y(\boldsymbol{x}^*)$.

A **dynamic min-max Stackelberg game**, $\left\{(X, Y, f^{(t)}, \boldsymbol{g}^{(t)})\right\}_{t=1}^T$, is a sequence of min-max Stackelberg games played for $T$ time periods. We define the players' value functions at time $t$ in a dynamic min-max Stackelberg game in the obvious way. Note that when $\boldsymbol{g}^{(t)}(\boldsymbol{x}, \boldsymbol{y}) \geq 0$ for all $\boldsymbol{x} \in X, \boldsymbol{y} \in Y$ and all time periods $t \in [T]$, the game reduces to a dynamic min-max game (with independent strategy sets). Moreover, if $\forall t, t' \in [T], f^{(t)} = f^{(t')}$, and $\boldsymbol{g}^{(t)} = \boldsymbol{g}^{(t')}$, then the game reduces to a (static) min-max Stackelberg game, which we denote simply by $(X, Y, f, \boldsymbol{g})$.

**Mathematical Preliminaries** Given $A \subset \mathbb{R}^n$, the function $f : A \to \mathbb{R}$ is said to be $\ell_f$-**Lipschitz-continuous** iff $\forall \boldsymbol{x}_1, \boldsymbol{x}_2 \in X$, $\|f(\boldsymbol{x}_1) - f(\boldsymbol{x}_2)\| \leq \ell_f \|\boldsymbol{x}_1 - \boldsymbol{x}_2\|$. If the gradient of $f$, $\nabla f$, is $\ell_{\nabla f}$-Lipschitz-continuous, we refer to $f$ as $\ell_{\nabla f}$-**Lipschitz-smooth**. We provide a review of online convex optimization in Appendix A.

# 3 No-Regret Learning Dynamics

In this section we explore no-regret learning dynamics in min-max Stackelberg games, and prove the convergence of no-regret learning dynamics in two settings: a pessimistic setting in which the outer player is a no-regret learner while the inner player best-responds, and an optimistic setting in which both players are no-regret learners. All the results in this paper rely on the following assumptions:

**Assumption 1.** *1. (*Slater's condition *(Slater 1959, 2014))* $\forall \boldsymbol{x} \in X, \exists \widehat{\boldsymbol{y}} \in Y$ *s.t.* $g_k(\boldsymbol{x}, \widehat{\boldsymbol{y}}) > 0$*, for all* $k = 1, \ldots, K$*; 2.* $f, g_1, \ldots, g_K$ *are continuous and convex-concave; and 3.* $\nabla_{\boldsymbol{x}} f, \nabla_{\boldsymbol{x}} g_1, \ldots, \nabla_{\boldsymbol{x}} g_K$ *are well-defined for all* $(\boldsymbol{x}, \boldsymbol{y}) \in X \times Y$ *and continuous in* $(\boldsymbol{x}, \boldsymbol{y})$*.*

We note that these assumptions are in line with previous work geared towards solving min-max Stackelberg games (Goktas and Greenwald 2021). Part 1 of Assumption 1, Slater's condition, is a standard constraint qualification condition (Boyd, Boyd, and Vandenberghe 2004),

which is needed to derive the optimality conditions for the inner player's maximization problem; without it the problem becomes analytically intractable. Part 2 of Assumption 1 is is required for the value function of the outer player to be continuous and convex ((Goktas and Greenwald 2021), Proposition A1) so that the problem is solvable efficiently. Finally, we note that Part 3 of Assumption 1 can be replaced by a subgradient boundedness assumption instead; however, for simplicity, we assume this stronger condition.

**Pessimistic Learning Setting**

In Stackelberg games, the leader decides their strategy assuming that the inner player will best respond which leads us to first consider a repeated game setting in which the inner player always best responds to the strategy picked by the outer player. Such a setting also makes sense as in zero-sum Stackelberg games the outer player and inner player are adversaries, and in most applications of interest we are concerned by optimal strategies for the outer player; hence, assuming a strong adversary which always best-responds allows us to consider more robust strategies for the outer player.

For any $\boldsymbol{x} \in X$, denote $\boldsymbol{y}^*(\boldsymbol{x}) \in \mathrm{BR}_Y(\boldsymbol{x})$, in such a setting, intuitively, the regret should be equal to the difference between the cumulative loss of the outer player w.r.t. to their sequence of actions to which the inner player best responds, and the smallest cumulative loss that the outer player can achieve by picking a fixed strategy to which the inner player best responds, i.e., $\frac{1}{T} \sum_{t=1}^T f^{(t)}(\boldsymbol{x}^{(t)}, \boldsymbol{y}^*(\boldsymbol{x}^{(t)})) - \sum_{t=1}^T \frac{1}{T} f^{(t)}(\boldsymbol{x}, \boldsymbol{y}^*(\boldsymbol{x}))$. We call this regret the **pessimistic regret** which can be more conveniently defined as the regret incurred by an action $\boldsymbol{x} \in X$ of the outer player w.r.t. a sequence of actions $\left\{(X, Y, f^{(t)}, \boldsymbol{g}^{(t)})\right\}_{t=1}^T$ and a dynamic min-max Stackelberg game $\{G_t\}_{t \in T}$ w.r.t. to the loss given by their value function $\{V_X^{(t)}\}_{t=1}^T$, i.e.:

$$\mathrm{PesRegret}_X^{(T)}(\boldsymbol{x}) = \frac{1}{T} \sum_{t=1}^T V_X^{(t)}(\boldsymbol{x}^{(t)}) - \sum_{t=1}^T \frac{1}{T} V_X^{(t)}(\boldsymbol{x})$$

(1)

That is, the pessimistic regret of the outer player compares the outer player's play history to the smallest cumulative loss the outer player could achieve by picking a fixed strategy *assuming that the inner player best-responds*. It is pessimistic in the sense that the outer player assumes the worst possible outcome for themself.

The main theorem in this section states the following: assuming the inner player best responds to the actions of the outer player, if the outer player employs a no-regret algorithm, then the outer player's average strategy converges to a Stackelberg equilibrium. Before presenting this theorem,[2] we recall the following property of the outer player's value function.

**Proposition 2** ((Goktas and Greenwald 2021), Proposition A.1)**.** *In a min-max Stackelberg game* $(X, Y, f, \boldsymbol{g})$*, the outer*

---

[2]The proofs of all mathematical claims in this section can be found in Appendix C.

player's value function, $V(\boldsymbol{x}) = \max_{\boldsymbol{y}\in Y:\boldsymbol{g}(\boldsymbol{x},\boldsymbol{y})\geq\mathbf{0}} f(\boldsymbol{x},\boldsymbol{y})$, is continuous and convex.

**Theorem 3.** *Consider a min-max Stackelberg game* $(X, Y, f, \boldsymbol{g})$, *and suppose the outer player plays a sequence of actions* $\{\boldsymbol{x}^{(t)}\}_{t=1}^T \subset X$. *If, after $T$ iterations, the outer player's pessimistic regret is bounded by $\varepsilon$ for all $\boldsymbol{x} \in X$, then* $(\bar{\boldsymbol{x}}^{(T)}, \boldsymbol{y}^*(\bar{\boldsymbol{x}}^{(T)}))$ *is a $(\varepsilon, 0)$-Stackelberg equilibrium, where* $\boldsymbol{y}^*(\bar{\boldsymbol{x}}^{(T)}) \in \mathrm{BR}_Y(\bar{\boldsymbol{x}}^{(T)})$.

We remark that even though the definition of pessimistic regret looks similar to the standard definition of regret, its structure is very different. In particular, without Proposition 2, it is not clear that the value $\sum_{t=1}^T \frac{1}{T} f^{(t)}(\boldsymbol{x}, \boldsymbol{y}^*(\boldsymbol{x})) = \sum_{t=1}^T V^{(t)}(\boldsymbol{x})$ is convex in $\boldsymbol{x}$.

## Optimistic Learning Setting

We now turn our attention to a learning setting in which both players are no-regret learners. The most straightforward way to define regret is by considering the outer and inner players' "vanilla" regrets, respectively: $\mathrm{Regret}_X^{(T)}(\boldsymbol{x}) = \frac{1}{T}\sum_{t=1}^T f(\boldsymbol{x}^{(t)}, \boldsymbol{y}^{(t)}) - \frac{1}{T}\sum_{t=1}^T f(\boldsymbol{x}, \boldsymbol{y}^{(t)})$ and $\mathrm{Regret}_Y^{(T)}(\boldsymbol{y}) = \frac{1}{T}\sum_{t=1}^T f(\boldsymbol{x}^{(t)}, \boldsymbol{y}) - \frac{1}{T}\sum_{t=1}^T f(\boldsymbol{x}^{(t)}, \boldsymbol{y}^{(t)})$. In convex-concave min-max games (with independent strategy sets), when both players minimize their vanilla regret, the players' average strategies converge to Nash equilibrium. In min-max Stackelberg games (with dependent strategy sets), however, convergence to a Stackelberg equilibrium in not guaranteed.

**Example 4.** *Consider the min-max Stackelberg game* $\min_{x\in[-1,1]} \max_{y\in[-1,1]:0\leq 1-(x+y)} x^2 + y + 1$. *The Stackelberg equilibrium of this game is given by $x^* = 1/2, y^* = 1/2$. Suppose both players employ no-regret algorithms that generate strategies $\{\boldsymbol{x}^{(t)}, \boldsymbol{y}^{(t)}\}_{t\in\mathbb{N}_+}$. Then at time $T \in \mathbb{N}_+$, there exists $\varepsilon > 0$, s.t.*

$$\begin{cases} \frac{1}{T}\sum_{t=1}^T \left[ x^{(t)^2} + y^{(t)} + 1 \right] - \frac{1}{T}\min_{x\in[-1,1]}\sum_{t=1}^T \left[ x^2 + y^{(t)} + 1 \right] \leq \varepsilon \\ \frac{1}{T}\max_{y\in[-1,1]}\sum_{t=1}^T \left[ x^{(t)^2} + y + 1 \right] - \frac{1}{T}\sum_{t=1}^T \left[ x^{(t)^2} + y^{(t)} + 1 \right] \leq \varepsilon \end{cases} \quad (2)$$

*Simplifying yields:*

$$\begin{cases} \frac{1}{T}\sum_{t=1}^T x^{(t)^2} - \min_{x\in[-1,1]} x^2 \leq \varepsilon \\ \max_{y\in[-1,1]} y - \frac{1}{T}\sum_{t=1}^T y^{(t)} \leq \varepsilon \end{cases} \quad (3)$$

*Since both players are no-regret learners, there exists $T \in \mathbb{N}_+$ large enough s.t.*

$$\begin{cases} \frac{1}{T}\sum_{t=1}^T x^{(t)^2} \leq \min_{x\in[-1,1]} x^2 \\ \max_{y\in[-1,1]} y \leq \frac{1}{T}\sum_{t=1}^T y^{(t)} \end{cases} = \begin{cases} \frac{1}{T}\sum_{t=1}^T x^{(t)^2} \leq 0 \\ 1 \leq \frac{1}{T}\sum_{t=1}^T y^{(t)} \end{cases} \quad (4)$$

*In other words, the average iterates converge to $x = 0, y = 1$, which is not the Stackelberg equilibrium of this game.*

If the inner player minimizes their vanilla regret without regard to the game's constraints, then their actions are not guaranteed to be feasible, and thus cannot converge to a

Stackelberg equilibrium. To remedy this infeasibility, we introduce a new type of regret we call **Lagrangian regret**, and show that assuming access to a solution oracle for the optimal KKT multipliers of the game's constraints, if both players minimize their Lagrangian regret, then no-regret learning dynamics converge to a Stackelberg equilibrium.

Define $\mathcal{L}_{\boldsymbol{x}}(\boldsymbol{y}, \boldsymbol{\lambda}) = f(\boldsymbol{x},\boldsymbol{y}) + \sum_{k=1}^K \lambda_k g_k(\boldsymbol{x},\boldsymbol{y})$ to be the Lagrangian associated with the outer player's value function, or equivalently, the inner player's maximization problem given the outer player's strategy $\boldsymbol{x} \in X$. If the optimal KKT multipliers $\boldsymbol{\lambda}^* \in \mathbb{R}^K$, which are guaranteed to exist by Slater's condition (Slater 1959), were known for the problem $\min_{\boldsymbol{x}\in X}\max_{\boldsymbol{y}\in Y:\boldsymbol{g}(\boldsymbol{x},\boldsymbol{y})\geq\mathbf{0}} f(\boldsymbol{x},\boldsymbol{y}) = \min_{\boldsymbol{x}\in X}\max_{\boldsymbol{y}\in Y}\min_{\boldsymbol{\lambda}\geq\mathbf{0}} \mathcal{L}_{\boldsymbol{x}}(\boldsymbol{y}, \boldsymbol{\lambda})$, then one could plug them back into the Lagrangian to obtain a convex-concave saddle point problem given by $\min_{\boldsymbol{x}\in X}\max_{\boldsymbol{y}\in Y} \mathcal{L}_{\boldsymbol{x}}(\boldsymbol{y}, \boldsymbol{\lambda}^*)$. Note that a saddle point of this problem is guaranteed to exist by the minimax theorem (Neumann 1928), since $\mathcal{L}_{\boldsymbol{x}}(\boldsymbol{y}, \boldsymbol{\lambda}^*)$ is convex in $\boldsymbol{x}$ and concave in $\boldsymbol{y}$. The next lemma states that the Stackelberg equilibria of a min-max Stackelberg game correspond to the saddle points of $\mathcal{L}_{\boldsymbol{x}}(\boldsymbol{y}, \boldsymbol{\lambda}^*)$.

**Lemma 5.** *Any Stackelberg equilibrium $(\boldsymbol{x}^*\boldsymbol{y}^*) \in X \times Y$ of any min-max Stackelberg game $(X, Y, f, \boldsymbol{g})$ corresponds to a saddle point of $\mathcal{L}_{\boldsymbol{x}}(\boldsymbol{y}, \boldsymbol{\lambda}^*)$, where $\boldsymbol{\lambda}^* \in \arg\min_{\boldsymbol{\lambda}\geq 0}\min_{\boldsymbol{x}\in X}\max_{\boldsymbol{y}\in Y} \mathcal{L}_{\boldsymbol{x}}(\boldsymbol{y}, \boldsymbol{\lambda})$.*

This lemma tells us that the function $\mathcal{L}_{\boldsymbol{x}}(\boldsymbol{y}, \boldsymbol{\lambda}^*)$ represents a new loss function that enforces the game's constraints. Based on this observation, we assume access to a Lagrangian solution oracle that provides us with $\boldsymbol{\lambda}^* \in \arg\min_{\boldsymbol{\lambda}\geq 0}\min_{\boldsymbol{x}\in X}\max_{\boldsymbol{y}\in Y} \mathcal{L}_{\boldsymbol{x}}(\boldsymbol{y}, \boldsymbol{\lambda}^*)$.

Further, we define a new type of regret which we call Lagrangian regret. Given a sequence of actions $\{\boldsymbol{x}^{(t)}, \boldsymbol{y}^{(t)}\}_{t=1}^T$ taken by the outer and inner players in a dynamic min-max Stackelberg game $\{(X, Y, f^{(t)}, \boldsymbol{g}^{(t)})\}_{t=1}^T$, we define their Lagrangian regret, respectively, as $\mathrm{LagrRegret}_X^{(T)}(\boldsymbol{x}) = \frac{1}{T}\sum_{t=1}^T \mathcal{L}_{\boldsymbol{x}^{(t)}}^{(t)}(\boldsymbol{y}^{(t)}, \boldsymbol{\lambda}^*) - \frac{1}{T}\sum_{t=1}^T \mathcal{L}_{\boldsymbol{x}}^{(t)}(\boldsymbol{y}^{(t)}, \boldsymbol{\lambda}^*)$ and $\mathrm{LagrRegret}_Y^{(T)}(\boldsymbol{y}) = \frac{1}{T}\sum_{t=1}^T \mathcal{L}_{\boldsymbol{x}^{(t)}}^{(t)}(\boldsymbol{y}, \boldsymbol{\lambda}^*) - \frac{1}{T}\sum_{t=1}^T \mathcal{L}_{\boldsymbol{x}^{(t)}}^{(t)}(\boldsymbol{y}^{(t)}, \boldsymbol{\lambda}^*)$.

The saddle point residual of a point $(\boldsymbol{x}^*, \boldsymbol{y}^*) \in X \times Y$ with respect to a convex-concave function $f : X \times Y \to \mathbb{R}$ is given by $\max_{\boldsymbol{y}\in Y} f(\boldsymbol{x}^*, \boldsymbol{y}) - \min_{\boldsymbol{x}\in X} f(\boldsymbol{x}, \boldsymbol{y}^*)$. When the saddle point residual is 0, the saddle point is a $(0,0)$-Stackelberg equilibrium.

The main theorem of this section now follows: if both players play so as to minimize their Lagrangian regret, then their average strategies converge to a Stackelberg equilibrium. The bound is given in terms of the saddle point residual of $\mathcal{L}_{\boldsymbol{x}}(\boldsymbol{y}, \boldsymbol{\lambda}^*)$.

**Theorem 6.** *Consider a min-max Stackelberg game $(X, Y, f, \boldsymbol{g})$, and suppose the outer and the players generate sequences of actions $\{(\boldsymbol{x}^{(t)}, \boldsymbol{y}^{(t)})\}_{t=1}^T \subset X$ using a no-Lagrangian-regret algorithm. If after $T$ iterations, the Lagrangian regret of both players is bounded by $\varepsilon$ for all*

$\boldsymbol{x} \in X$, the following convergence bound holds on the saddle point residual of $(\bar{\boldsymbol{x}}^{(T)}, \bar{\boldsymbol{y}}^{(T)})$ w.r.t. the Lagrangian: $0 \leq \max_{\boldsymbol{y} \in Y} \mathcal{L}_{\bar{\boldsymbol{x}}^{(T)}}(\boldsymbol{y}, \boldsymbol{\lambda}^*) - \min_{\boldsymbol{x} \in X} \mathcal{L}_{\boldsymbol{x}}(\bar{\boldsymbol{y}}^{(T)}, \boldsymbol{\lambda}^*) \leq 2\varepsilon$.

Having provided convergence to Stackelberg equilibrium of general no-regret learning dynamics in min-max Stackelberg games, we now proceed to investigate the convergence and robustness properties of a specific example of a no-regret learning dynamic, namely online mirror descent (OMD) dynamics.

## 4 Online Mirror Descent

In this section, we apply the results we have derived for no-regret learning dynamics to Online Mirror Descent (OMD) (Zinkevich 2003; Shalev-Shwartz et al. 2011). We apply the theorems we derived above to OMD, and then we study the robustness properties of OMD in min-max Stackelberg games.

**Convergence Analysis**

When the outer player is an OMD learner minimizing its pessimistic regret and the inner player best responds, we obtain the max-oracle gradient descent algorithm (Algorithm 1 - Appendix D) first proposed by Jin, Netrapalli, and Jordan (2020) for min-max games.

Following Jin, Netrapalli, and Jordan (2020), Goktas and Greenwald extend the max-oracle gradient descent algorithm to min-max Stackelberg games and prove its convergence of in best iterates. The following corollary of Theorem 3, which concerns convergence of this algorithm in average iterates, complements their result: the max-oracle gradient descent algorithm is guaranteed to converge to an $(\varepsilon, 0)$-Stackelberg equilibrium strategy of the outer player in average iterates after $O(1/\varepsilon^2)$ iterations, assuming the inner player best responds.

We note that since $V_X$ is convex, by Proposition 2, $V_X$ is subdifferentiable. Moreover, for all $\widehat{\boldsymbol{x}} \in X$, $\widehat{\boldsymbol{y}} \in \mathrm{BR}_Y(\widehat{\boldsymbol{x}})$, $\nabla_{\boldsymbol{x}} f(\widehat{\boldsymbol{x}}, \widehat{\boldsymbol{y}}) + \sum_{k=1}^{K} \lambda_k g_k(\widehat{\boldsymbol{x}}, \widehat{\boldsymbol{y}})$ is an arbitrary subgradient of the value function at $\widehat{\boldsymbol{x}}$ by Goktas and Greenwald's subdifferential envelope theorem (2021). We add that similar to Goktas and Greenwald, we assume that the optimal KKT multipliers $\boldsymbol{\lambda}^*(\boldsymbol{x}^{(t)}, \widehat{\boldsymbol{y}}(\boldsymbol{x}^{(t)}))$ associated with a solution $\widehat{\boldsymbol{y}}(\boldsymbol{x}^{(t)}))$ can be computed in constant time.

**Corollary 7.** *Let* $c = \max_{\boldsymbol{x} \in X} \|\boldsymbol{x}\|$ *and let* $\ell_f = \max_{(\widehat{\boldsymbol{x}}, \widehat{\boldsymbol{y}}) \in X \times Y}$ $\|\nabla_{\boldsymbol{x}} f(\widehat{\boldsymbol{x}}, \widehat{\boldsymbol{y}})\|$. *If Algorithm 1 (Appendix D) is run on a min-max Stackelberg game* $(X, Y, f, \boldsymbol{g})$ *with* $\eta_t = \frac{c}{\ell_f \sqrt{2T}}$ *for all iteration* $t \in [T]$ *and any* $\boldsymbol{x}^{(0)} \in X$, *then* $(\bar{\boldsymbol{x}}^{(T)}, \boldsymbol{y}^*(\bar{\boldsymbol{x}}^{(T)}))$ *is a* $(c\ell_f \sqrt{2}/\sqrt{T}, 0)$-*Stackelberg equilibrium. Furthermore, for* $\varepsilon \in (0, 1)$, *if we choose* $T \geq N_T(\varepsilon) \in O(1/\varepsilon^2)$, *then there exists an iteration* $T^* \leq T$ *s.t.* $(\bar{\boldsymbol{x}}^{(T)}, \boldsymbol{y}^*(\bar{\boldsymbol{x}}^{(T)}))$ *is an* $(\varepsilon, 0)$-*Stackelberg equilibrium.*

Note that we can relax Theorem 3 to instead work with an approximate best response of the inner player, i.e., given the strategy of the outer player $\widehat{\boldsymbol{x}}$, instead of playing an exact best-response, the inner player computes a $\widehat{\boldsymbol{y}}$ s.t. $f(\widehat{\boldsymbol{x}}, \widehat{\boldsymbol{y}}) \geq \max_{\boldsymbol{y} \in Y : \boldsymbol{g}(\widehat{\boldsymbol{x}}, \boldsymbol{y}) \geq \boldsymbol{0}} f(\widehat{\boldsymbol{x}}) - \varepsilon$. Combine with results on the

convergence of gradient ascent on smooth functions, the average iterates computed by Goktas and Greenwald's nested GDA algorithm converge to an $(\varepsilon, \varepsilon)$-Stackelberg equilibrium in $O(1/\varepsilon^3)$ iterations. If additionally, $f$ is strongly convex in $\boldsymbol{y}$, then the iteration complexity can reduced to $O(1/\varepsilon^2 \log(1/\varepsilon))$.

Similarly, we can also consider the optimistic case, in which both the outer and inner players minimize their Lagrangian regrets, as OMD learners with access to a Lagrangian solution oracle that returns $\boldsymbol{\lambda}^* \in \arg\min_{\boldsymbol{\lambda} \geq 0} \min_{\boldsymbol{x} \in X} \max_{\boldsymbol{y} \in Y} \mathcal{L}_{\boldsymbol{x}}(\boldsymbol{y}, \boldsymbol{\lambda}^*)$. In this case, we obtain the Lagrangian GDA (LGDA) algorithm (Algorithm 2 - Appendix D). The following corollary of Theorem 6 states that LGDA converges in average iterates to an approximate-Stackelberg equilibrium in $O(1/\varepsilon^2)$ iterations.

**Corollary 8.** *Let* $b = \max_{\boldsymbol{x} \in X} \|\boldsymbol{x}\|$, $c = \max_{\boldsymbol{y} \in Y} \|\boldsymbol{y}\|$, *and* $\ell_{\mathcal{L}} = \max_{(\widehat{\boldsymbol{x}}, \widehat{\boldsymbol{y}}) \in X \times Y} \|\nabla_{\boldsymbol{x}} \mathcal{L}_{\widehat{\boldsymbol{x}}}(\widehat{\boldsymbol{y}}, \boldsymbol{\lambda}^*)\|$. *If Algorithm 2 (Appendix D) is run on a min-max Stackelberg game* $(X, Y, f, \boldsymbol{g})$ *with* $\eta_t^{\boldsymbol{x}} = \frac{b}{\ell_{\mathcal{L}} \sqrt{2T}}$ *and* $\eta_t^{\boldsymbol{y}} = \frac{c}{\ell_{\mathcal{L}} \sqrt{2T}}$ *for all iterations* $t \in [T]$ *and any* $\boldsymbol{x}^{(0)} \in X$, *then the following convergence bound holds on the saddle point residual* $(\bar{\boldsymbol{x}}^{(T)}, \bar{\boldsymbol{y}}^{(T)})$ *w.r.t. the Lagrangian:*

$$0 \leq \max_{\boldsymbol{y} \in Y} \mathcal{L}_{\bar{\boldsymbol{x}}^{(T)}}(\boldsymbol{y}, \boldsymbol{\lambda}^*) - \min_{\boldsymbol{x} \in X} \mathcal{L}_{\boldsymbol{x}}(\bar{\boldsymbol{y}}^{(T)}, \boldsymbol{\lambda}^*)$$
$$\leq \frac{2\sqrt{2}\ell_{\mathcal{L}}}{\sqrt{T}} \max\{b, c\} \quad (5)$$

We remark that in certain rare cases the Lagrangian can become degenerate in $\boldsymbol{y}$, in that the $\boldsymbol{y}$ terms in the Lagrangian might cancel out when $\boldsymbol{\lambda}^*$ is plugged back into Lagrangian, leading LGDA to not update the $\boldsymbol{y}$ variables, as demonstrated by the following example:

**Example 9.** *Consider this min-max Stackelberg game:* $\min_{x \in [-1, 1]}$ $\max_{y \in [-1, 1] : 0 \leq 1 - (x+y)} x^2 + y + 1$. *When we plug the optimal KKT multiplier* $\lambda^* = 1$ *into the Lagrangian associated with the outer player's value function, we obtain* $\mathcal{L}_x(y, \lambda) = x^2 + y + 1 - (x + y) = x^2 - x + 1$, *with* $\frac{\partial \mathcal{L}}{\partial x} = 2x - 1$ *and* $\frac{\partial \mathcal{L}}{\partial y} = 0$. *It follows that the* $x$ *iterate converges to* $1/2$, *but the* $y$ *iterate will never be updated, and hence unless* $y$ *is initialized to its Stackelberg equilibirium value, LGDA will not converge to a Stackelberg equilibrium.*

In general, this degeneracy issue occurs when $\forall \boldsymbol{x} \in X, \nabla_{\boldsymbol{y}} f(\boldsymbol{x}, \boldsymbol{y}) = -\sum_{k=1}^{K} \lambda_k^* \nabla_{\boldsymbol{y}} g_k(\boldsymbol{x}, \boldsymbol{y})$. We can sidestep the issue by restricting our attention to min-max Stackelberg games with convex-*strictly*-concave objective functions, which is *sufficient* to ensure that the Lagrangian is not degenerate in $\boldsymbol{y}$ (Boyd, Boyd, and Vandenberghe 2004).

**Robustness Analysis**

Although the OMD dynamics we analyzed in the previous section describe a dynamic behavior in nature, they assume that the game and its properties, i.e., the objective function and constraints, are static and thus do not change over time. In many real-world games, however, the game itself is subject to perturbations, i.e., dynamic changes, in the sense that

the agents' objectives and constraints might be perturbed by external influences. Analyzing and providing dynamics that are robust to ongoing changes in games is critical, since the real world is rarely static.

This makes the study of dynamic min-max Stackelberg games and their associated optimal dynamic strategies for both players an important goal. Dynamic games bring with them a series of interesting issues; notably, even though the environment might change at each time period, in each period of time the game still exhibits a Stackelberg equilibrium. However, one cannot sensibly expect the players to play a Stackelberg equilibrium strategy at each time period since even in the static setting, known game dynamics require multiple time steps in order for players to reach even an approximate Stackelberg equilibrium. When players cannot directly best respond or pick the optimal strategy for themselves, they essentially become boundedly rational agents in that they can only take a step towards their optimal strategy but they cannot reach it in just one time step. Hence, in dynamic games, equilibria also become dynamic objects, which can never be reached unless the game stops changing significantly.

Corollaries 8 and 7 tell us that OMD dynamics are effective equilibrium finding strategies in min-max Stackelberg games. However, they do not provide any intuition about the robustness of OMD dynamics to perturbations in the game. That is, we would like to know whether or not OMD dynamics are able to track the equilibrium even when the game changes slowly. Robustness is a desirable property for no-regret learning dynamics as many real-world applications of games involve changing environments. In this section, we provide theoretical guarantees that show that even when the game changes at each iteration, OMD dynamics closely track the changing equilibria of the dynamic game. Unfortunately, our theoretical results only concern min-max games (with independent strategy sets). Nevertheless, we provide experimental evidence that suggests that the results we prove may also apply more broadly to min-max Stackelberg games (with dependent strategy sets).

We first consider the pessimistic setting in which the outer player is a no-regret learner and the inner player best-responds. In this setting, we show that when the outer player follows online projected gradient descent dynamics in a dynamic min-max game, i.e., a min-max game in which the objective function constantly changes, the outer player's strategies closely track their Stackelberg equilibrium strategy. Intuitively, the following result implies that irrespective of the initial strategy of the outer player, online projected gradient descent dynamics follow the Nash equilibrium strategy of the outer player s.t. the strategy determined by the outer player is always within a $2d/\delta$ radius of the outer player's Nash equilibrium strategy.

**Theorem 10.** *Consider a dynamic min-max game $\left\{(X, Y, f^{(t)})\right\}_{t=1}^{T}$. Suppose that, for all $t \in [T]$, $f^{(t)}$ is $\mu$-strongly convex in $\boldsymbol{x}$ and strictly concave in $\boldsymbol{y}$, and $f^{(t)}$ is $\ell_{\nabla f}$-Lipschitz smooth. Suppose that the outer player generates a sequence of actions $\{\boldsymbol{x}^{(t)}\}_{t=1}^{T} \subset X$ by using an online projected gradient descent algorithm on the loss*

*functions $\{V^{(t)}\}_{t=1}^{T}$ with learning rate $\eta \leq \frac{2}{\mu + \ell_{\nabla f}}$ and suppose that the inner player generates a sequence of best-responses to each iterate of the outer player $\{\boldsymbol{y}^{(t)}\}_{t=1}^{T} \subset Y$. For all $t \in [T]$, let $\boldsymbol{x}^{(t)^*} \in \arg\min_{\boldsymbol{x} \in X} V^{(t)}(\boldsymbol{x})$, $\Delta^{(t)} = \left\|\boldsymbol{x}^{(t+1)^*} - \boldsymbol{x}^{(t)^*}\right\|$, and $\delta = \frac{2\eta\mu\ell_{\nabla f}}{\ell_{\nabla f} + \mu}$, we then have:*

$$\left\|\boldsymbol{x}^{(T)^*} - \boldsymbol{x}^{(T)}\right\| \leq (1-\delta)^{T/2} \left\|\boldsymbol{x}^{(0)^*} - \boldsymbol{x}^{(0)}\right\|$$
$$+ \sum_{t=1}^{T} (1-\delta)^{\frac{T-t}{2}} \Delta^{(t)} \quad (6)$$

*If additionally, for all $t \in [T]$, $\Delta^{(t)} \leq d$, then:*

$$\left\|\boldsymbol{x}^{(T)^*} - \boldsymbol{x}^{(T)}\right\| \leq (1-\delta)^{T/2} \left\|\boldsymbol{x}^{(0)^*} - \boldsymbol{x}^{(0)}\right\| + \frac{2d}{\delta} \quad (7)$$

We can extend a similar robustness result to the setting in which the outer and inner players are both OMD learners. The following theorem implies that irrespective of the initial strategies of the two players, online projected gradient descent dynamics follow the Nash equilibrium of the game, always staying within a $4d/\delta$ radius.

**Theorem 11.** *Consider a dynamic min-max game $\{G_t\}_{t=0}^{T} = \left\{(X, Y, f^{(t)})\right\}_{t=1}^{T}$. Suppose that, for all $t \in [T]$, $f^{(t)}$ is $\mu_{\boldsymbol{x}}$-strongly convex in $\boldsymbol{x}$ and $\mu_{\boldsymbol{y}}$-strongly concave in $\boldsymbol{y}$, $f^{(t)}$ is $\ell_{\nabla f}$-Lipschitz smooth. Let $\{(\boldsymbol{x}^{(t)}, \boldsymbol{y}^{(t)})\}_{t=1}^{T} \subset X \times Y$ be the strategies generated by the outer and inner players assuming that the outer player uses a online projected gradient descent algorithm on the losses $\{f^{(t)}(\cdot, \boldsymbol{y}^{(t)})\}_{t=1}^{T}$ with $\eta_{\boldsymbol{x}} = \frac{2}{\mu_{\boldsymbol{x}} + \ell_{\nabla f}}$ and that the inner player uses a online projected gradient descent algorithm on the losses $\{-f^{(t)}(\boldsymbol{x}^{(t)}, \cdot)\}_{t=1}^{T}$ with $\eta_{\boldsymbol{y}} = \frac{2}{\mu_{\boldsymbol{y}} + \ell_{\nabla f}}$. For all $t \in [T]$, let $\boldsymbol{x}^{(t)^*} \in \arg\min_{\boldsymbol{x} \in X} f^{(t)}(\boldsymbol{x}, \boldsymbol{y}^{(t)})$, $\boldsymbol{y}^{(t)^*} \in \arg\min_{\boldsymbol{y} \in Y} f^{(t)}(\boldsymbol{x}^{(t)}, \boldsymbol{y})$. $\Delta_{\boldsymbol{x}}^{(t)} = \left\|\boldsymbol{x}^{(t+1)^*} - \boldsymbol{x}^{(t)^*}\right\|$, $\Delta_{\boldsymbol{y}}^{(t)} = \left\|\boldsymbol{y}^{(t+1)^*} - \boldsymbol{y}^{(t)^*}\right\|$, $\delta_{\boldsymbol{x}} = \frac{2\eta\mu_{\boldsymbol{x}}\ell_{\nabla f}}{\ell_{\nabla_{\boldsymbol{x}} f} + \mu_{\boldsymbol{x}}}$, and $\delta_{\boldsymbol{y}} = \frac{2\eta\mu_{\boldsymbol{y}}\ell_{\nabla f}}{\ell_{\nabla f} + \mu_{\boldsymbol{y}}}$ we then have:*

$$\left\|\boldsymbol{x}^{(T)^*} - \boldsymbol{x}^{(T)}\right\| + \left\|\boldsymbol{y}^{(T)^*} - \boldsymbol{y}^{(T)}\right\|$$
$$\leq (1-\delta_{\boldsymbol{x}})^{T/2} \left\|\boldsymbol{x}^{(0)^*} - \boldsymbol{x}^{(0)}\right\| + (1-\delta_{\boldsymbol{y}})^{T/2} \left\|\boldsymbol{y}^{(0)^*} - \boldsymbol{y}^{(0)}\right\|$$
$$+ \sum_{t=1}^{T} (1-\delta_{\boldsymbol{x}})^{\frac{T-t}{2}} \Delta_{\boldsymbol{x}}^{(t)} + \sum_{t=1}^{T} (1-\delta_{\boldsymbol{y}})^{\frac{T-t}{2}} \Delta_{\boldsymbol{y}}^{(t)} \ . \quad (8)$$

*If additionally, $\Delta_{\boldsymbol{x}}^{(t)} \leq d$ and $\Delta_{\boldsymbol{y}}^{(t)} \leq d$ for all $t \in [T]$, and $\delta = \min\{\delta_{\boldsymbol{y}}, \delta_{\boldsymbol{x}}\}$, then:*

$$\left\|\boldsymbol{x}^{(T)^*} - \boldsymbol{x}^{(T)}\right\| + \left\|\boldsymbol{y}^{(T)^*} - \boldsymbol{y}^{(T)}\right\|$$
$$\leq 2(1-\delta)^{T/2} \left(\left\|\boldsymbol{x}^{(0)^*} - \boldsymbol{x}^{(0)}\right\| + \left\|\boldsymbol{y}^{(0)^*} - \boldsymbol{y}^{(0)}\right\|\right) + \frac{4d}{\delta} \ . \quad (9)$$

The proofs of the above theorems are relegated to Appendix C. The theorems we have proven in this section establish the robustness of OMD dynamics for min-max games in both the pessimistic and optimistic settings by showing that the dynamics closely track the Stackelberg equilibrium in a large class of min-max games. As we are not able to extend these theoretical robustness guarantees to min-max Stackelberg games (with dependent strategy sets), we instead ran a series of experiments on (dynamic) Fisher markets, which are canonical examples of min-max Stackelberg games (Goktas and Greenwald 2021), to investigate the empirical robustness guarantees of OMD dynamics for this class of min-max Stackelberg games.

## Dynamic Fisher Markets

The Fisher market model, attributed to Irving Fisher (Brainard, Scarf et al. 2000), has received a great deal of attention in the literature, especially by computer scientists, as it has proven useful in the design of online marketplaces. We now study OMD dynamics in dynamic Fisher markets, which are instances of min-max Stackelberg games (Goktas and Greenwald 2021).

A **Fisher market** consists of $n$ buyers and $m$ divisible goods (Brainard, Scarf et al. 2000). Each buyer $i \in [n]$ has a budget $b_i \in \mathbb{R}_+$ and a utility function $u_i : \mathbb{R}_+^m \to \mathbb{R}$. Each good $j \in [m]$ has supply $s_j \in \mathbb{R}_+$. A Fisher market is thus given by a tuple $(n, m, U, \boldsymbol{b}, \boldsymbol{s})$, where $U = \{u_1, \ldots, u_n\}$ is a set of utility functions, one per buyer, $\boldsymbol{b} \in \mathbb{R}_+^n$ is a vector of buyer budgets, and $\boldsymbol{s} \in \mathbb{R}_+^m$ is a vector of good supplies. We abbreviate as $(U, \boldsymbol{b}, \boldsymbol{s})$ when $n$ and $m$ are clear from context. A **dynamic Fisher market** is a sequence of Fisher markets $\left(U^{(t)}, \boldsymbol{b}^{(t)}, \boldsymbol{s}^{(t)}\right)_{t=1}^{([T])}$. An **allocation** $\boldsymbol{X} = (\boldsymbol{x}_1, \ldots, \boldsymbol{x}_n)^T \in \mathbb{R}_+^{n \times m}$ is a map from goods to buyers, represented as a matrix s.t. $x_{ij} \geq 0$ denotes the amount of good $j \in [m]$ allocated to buyer $i \in [n]$. Goods are assigned **prices** $\boldsymbol{p} = (p_1, \ldots, p_m)^T \in \mathbb{R}_+^m$. A tuple $(\boldsymbol{p}^*, \boldsymbol{X}^*)$ is said to be a **competitive (or Walrasian) equilibrium** of Fisher market $(U, \boldsymbol{b}, \boldsymbol{s})$ if 1. buyers are utility maximizing, constrained by their budget, i.e., $\forall i \in [n], \boldsymbol{x}_i^* \in \arg\max_{\boldsymbol{x}:\boldsymbol{x}\cdot\boldsymbol{p}^* \leq b_i} u_i(\boldsymbol{x})$; and 2. the market clears, i.e., $\forall j \in [m], p_j^* > 0 \Rightarrow \sum_{i \in [n]} x_{ij}^* = s_j$ and $p_j^* = 0 \Rightarrow \sum_{i \in [n]} x_{ij}^* \leq s_j$.

Goktas and Greenwald (2021) observe that any competitive equilibrium $(\boldsymbol{p}^*, \boldsymbol{X}^*)$ of a Fisher market $(U, \boldsymbol{b})$ corresponds to a Stackelberg equilibrium of the following min-max Stackelberg game:[3]

$$\min_{\boldsymbol{p} \in \mathbb{R}_+^m} \max_{\boldsymbol{X} \in \mathbb{R}_+^{n \times m}:\boldsymbol{X}\boldsymbol{p} \leq \boldsymbol{b}} \sum_{j \in [m]} s_j p_j + \sum_{i \in [n]} b_i \log(u_i(\boldsymbol{x}_i)) \ . \tag{10}$$

Let $\mathcal{L} : \mathbb{R}_+^m \times \mathbb{R}^{n \times m} \to \mathbb{R}_+$ be the Lagrangian of the outer player's value function in Equation (10), i.e.,
$\mathcal{L}_{\boldsymbol{p}}(\boldsymbol{X}, \boldsymbol{\lambda}) = \sum_{j \in [m]} s_j p_j + \sum_{i \in [n]} b_i \log(u_i(\boldsymbol{x}_i)) +$

---

[3]The first term in this program is slightly different than the first term in the program presented by Goktas and Greenwald (2021), since supply is assumed to be 1 their work.

$\sum_{i \in [n]} \lambda_i (b_i - \boldsymbol{x}_i \cdot \boldsymbol{p})$. One can show the existence of a Lagrangian solution oracle for the Lagrangian of Equation (10) such that $\boldsymbol{\lambda}^* = \boldsymbol{1}_m$. We then have: 1. by Goktas and Greenwald's envelope theorem, the subdifferential of the outer player's value function is given by $\nabla_{\boldsymbol{p}} V(\boldsymbol{p}) = \boldsymbol{s} - \sum_{i \in [n]} \boldsymbol{x}_i^*(\boldsymbol{p})$, where $\boldsymbol{x}_i^*(\boldsymbol{p}) \in \arg\max_{\boldsymbol{x} \in \mathbb{R}_+^m \boldsymbol{x}\cdot\boldsymbol{p} \leq b_i} u_i(\boldsymbol{x})$, 2. the gradient of the Lagrangian w.r.t. the prices, given the Langrangian solution oracle, is $\nabla_{\boldsymbol{p}} \mathcal{L}_{\boldsymbol{p}}(\boldsymbol{X}, \boldsymbol{\lambda}^*) = \boldsymbol{s} - \sum_{i \in [n]} \boldsymbol{x}_i$ and $\nabla_{\boldsymbol{x}_i} \mathcal{L}_{\boldsymbol{p}}(\boldsymbol{X}, \boldsymbol{\lambda}^*)) = \frac{b_i}{u_i(\boldsymbol{x}_i)} \nabla_{\boldsymbol{x}_i} u_i(\boldsymbol{x}_i) - \boldsymbol{p}$, where $\boldsymbol{\lambda}^* = \boldsymbol{1}_m$.

We first consider OMD dynamics for Fisher markets in the pessimistic setting, in which the outer player determines their strategy via online projected gradient descent and the inner player best-responds. In this setting, we obtain a dynamic version of a natural price adjustment process known as tâtonnement (Walras 1969), which was first studied by Cheung, Hoefer, and Nakhe (2019) (Algorithm 3, Appendix D).

We then consider OMD dynamics in the optimistic setting, in which case both the outer and inner players employ online projected gradient descent, which yields myopic best-response dynamics (Monderer and Shapley 1996) (Algorithm 4, Appendix D). In words, at each time step, the (fictional Walrasian) auctioneer takes a gradient descent step to minimize its regret, and then all the buyers take a gradient ascent step to minimize their Lagrangian regret. These gradient descent-ascent dynamics can be seen as myopic best-response dynamics for sellers and buyers who are both boundedly rational (Camerer 1998).

**Experiments** In order to better understand the robustness properties of Algorithms 3 and 4 in a dynamic min-max Stackelberg game that is subject to perturbation across time, we ran a series of experiments with dynamic Fisher Markets assuming three different classes of utility functions.[4] Each utility structure endows Equation (10) with different smoothness properties, which allows us to compare the efficiency of the algorithms under varying conditions. Let $\boldsymbol{v}_i \in \mathbb{R}^m$ be a vector of valuation parameters that describes the utility function of buyer $i \in [n]$. We consider the following utility function classes: 1. linear: $u_i(\boldsymbol{x}_i) = \sum_{j \in [m]} v_{ij} x_{ij}$; 2. Cobb-Douglas: $u_i(\boldsymbol{x}_i) = \prod_{j \in [m]} x_{ij}^{v_{ij}}$; and 3. Leontief: $u_i(\boldsymbol{x}_i) = \min_{j \in [m]} \left\{\frac{x_{ij}}{v_{ij}}\right\}$. To simulate the dynamic Fisher markets, we fix a range for every market parameter and draw from that range uniformly at random during each iteration. Our goal is to understand how closely OMD dynamics track the Stackelberg equilibria of the game as the latter varies with time. To do so, we compare the distance between the iterates $\left(\boldsymbol{p}^{(t)}, \boldsymbol{X}^{(t)}\right)$ computed by the algorithms and the equilibrium of the game at each iteration $t$. This distance is measured as $\left\|\boldsymbol{p}^{(t)^*} - \boldsymbol{p}^{(t)}\right\|_2 + \left\|\boldsymbol{X}^{(t)^*} - \boldsymbol{X}^{(t)}\right\|_2$, where $\left(\boldsymbol{p}^{(t)^*}, \boldsymbol{X}^{(t)^*}\right)$ is the Stackelberg equilibrium of the Fisher market $(U^{(t)}, \boldsymbol{b}^{(t)}, \boldsymbol{s}^{(t)})$ at time $t \in [T]$.

In our experiments, we ran Algorithms 3 and 4 on 100 ran-

---

[4]Our code can be found at https://anonymous.4open.science/r/Dynamic-Minmax-Games-8153/.

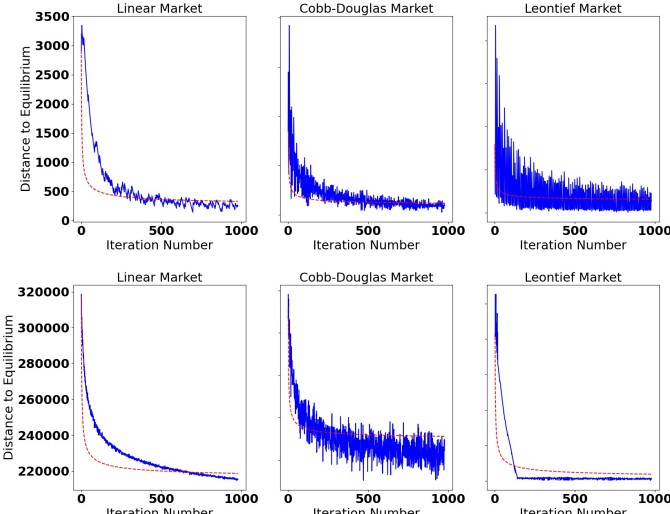

Figure 1: In blue, we depict a trajectory of distances between computed allocation-price pairs and equilibrium allocation-price pairs, when Algorithm 3 is run on randomly initialized dynamic linear, Cobb-Douglas, and Leontief Fisher markets. In red, we plot an arbitrary $O(1/\sqrt{T})$ function.

Figure 2: In blue, we depict a trajectory of distances between computed allocation-price pairs and equilibrium allocation-price pairs, when Algorithm 4 is run on randomly initialized dynamic linear, Cobb-Douglas, and Leontief Fisher markets. In red, we plot an arbitrary $O(1/\sqrt{T})$ function.

domly initialized dynamic Fisher markets. We depict the distance to equilibrium at each iteration for a randomly chosen experiment in Figures 1 and 2. In these figures, we observe that our OMD dynamics are closely tracking the Stackelberg equilibria as they vary with each iteration. A more detailed description of our experimental setup can be found in Appendix E.

We observe from Figures 1 and 2 that for both Algorithms 3 and 4, we obtain an empirical convergence rate relatively close to $O(1/\sqrt{T})$ under Cobb-Douglas utilities, and a slightly slower empirical convergence rate under linear utilities. Recall that $O(1/\sqrt{T})$ is the convergence rate guarantee we obtained for both algorithms, assuming a fixed learning rate in a static Fisher market (Corollaries 7 and 8).

Dynamic Fisher markets with Leontief utilities, in which the objective function is not differentiable, are the hardest markets of the three for our algorithms to solve. Still, we only see a slightly slower than $O(1/\sqrt{T})$ empirical convergence rate for both Algorithms 3 and 4. In these experiments, the convergence curve generated by Algorithm 4 has a less erratic behavior than the one generated by Algorithm 3. Due to the non-differentiability of the objective function, the gradient ascent step in Algorithm 4 for buyers with Leontief utilities is very small, effectively dampening any potentially erratic changes it the iterates.

Our experiments suggest that even when the game changes at each iteration, OMD dynamics (Algorithms 3 and 4 - Appendix D) are robust enough to closely track the changing Stackelberg equilibria of dynamic Fisher markets. We note that tâtonnement dynamics (Algorithm 3) seem to be more robust than myopic best response dynamics (Algorithm 4), i.e., the distance to equilibrium allocations is smaller at each iteration of tâtonnement. This result is not surprising, as tâtonnement computes a utility-maximizing allocation for the buyers at each time step. Even though Theorems 10 and 11 only provide theoretical guarantees on the robustness of OMD dynamics in dynamic min-max games (with independent strategy sets), it seems like similar theoretical robustness results may be attainable in dynamic min-

max Stackelberg games (with dependent strategy sets).

## 5 Conclusion

We began this paper by considering no-regret learning dynamics for min-max Stackelberg games in two settings: a pessimistic setting in which the outer player is a no-regret learner and the inner player best responds, and an optimistic setting in which both players are no-regret learners. For both of these settings, we proved that no-regret learning dynamics converge to a Stackelberg equilibrium of the game. We then specialized the no-regret algorithm employed by the players to online mirror descent (OMD), which yielded two known algorithms, namely max-oracle gradient descent (Jin, Netrapalli, and Jordan 2020) and nested GDA (Goktas and Greenwald 2021) in the pessimistic setting, and a new simultaneous GDA-like algorithm (Nedic and Ozdaglar 2009), which we call Lagrangian GDA, in the optimistic setting. As these algorithms are no-regret learning algorithms, our previous theorems imply convergence to Stackelberg equilibria in $O(1/\varepsilon^2)$ iterations. Finally, we investigated the robustness of OMD dynamics to perturbations in the parameters of a min-max Stackelberg game. To do so, we analyzed how closely OMD dynamics track Stackelberg equilibria in dynamic min-max Stackelberg games. We proved that in min-max games (with independent strategy sets) OMD dynamics closely track the changing Stackelberg equilibria of a game. As we were not able to extend these theoretical robustness guarantees to min-max Stackelberg games (with dependent strategy sets), we instead ran a series of experiments on dynamic Fisher markets, which are canonical examples of min-max Stackelberg games. Our experiments suggest that OMD dynamics are robust for min-max Stackelberg games so that perhaps the robustness guarantees we have provided for OMD dynamics in min-max games can be extended to min-max Stackelberg games. The theory developed in this paper opens the door to extending the myriad applications of Stackelberg games in AI to incorporating dependent strategy sets.

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

# A  Background

**Notation**  We use Roman uppercase letters to denote sets (e.g., $X$), bold uppercase letters to denote matrices (e.g., $\boldsymbol{X}$), bold lowercase letters to denote vectors (e.g., $\boldsymbol{p}$), and Roman lowercase letters to denote scalar quantities, (e.g., $c$). We denote the $i$th row vector of a matrix (e.g., $\boldsymbol{X}$) by the corresponding bold lowercase letter with subscript $i$ (e.g., $\boldsymbol{x}_i$). Similarly, we denote the $j$th entry of a vector (e.g., $\boldsymbol{p}$ or $\boldsymbol{x}_i$) by the corresponding Roman lowercase letter with subscript $j$ (e.g., $p_j$ or $x_{ij}$). We denote the vector of ones of size $n$ by $\boldsymbol{1}_n$. We denote the set of integers $\{1, \ldots, n\}$ by $[n]$, the set of natural numbers by $\mathbb{N}$, the set of positive natural numbers by $\mathbb{N}_+$ the set of real numbers by $\mathbb{R}$, the set of non-negative real numbers by $\mathbb{R}_+$, and the set of strictly positive real numbers by $\mathbb{R}_{++}$. We denote the orthogonal projection operator onto a convex set $C$ by $\Pi_C$, i.e., $\Pi_C(\boldsymbol{x}) = \arg\min_{\boldsymbol{y} \in C} \|\boldsymbol{x} - \boldsymbol{y}\|^2$. Given a sequence of iterates $\{\boldsymbol{z}^{(t)}\}_{t=1}^T \subset Z$, we denote the average iterate $\bar{\boldsymbol{z}}^{(T)} = \frac{1}{T} \sum_{t=1}^T \boldsymbol{z}^{(t)}$.

**Online Convex Optimization**  An **online convex optimization problem (OCP)** is a decision problem in a dynamic environment which comprises a finite time horizon $T$, a compact, convex feasible set $X$, and a sequence of convex differentiable loss functions $\{\ell^{(t)}\}_{t=1}^T$, where $\ell^{(t)} : X \to \mathbb{R}$ for all $t \in [T]$. A solution to an OCP is a sequence $\{\boldsymbol{x}^{(t)}\}_{t=1}^T \subset X$. A preferred solution is one that minimizes **average regret** given by $\mathrm{Regret}^{(T)}(\boldsymbol{x}) = \sum_{t=1}^T \frac{1}{T}\ell^{(t)}(\boldsymbol{x}^{(t)}) - \sum_{t=1}^T \frac{1}{T}\ell^{(t)}(\boldsymbol{x})$, over all $\boldsymbol{x} \in X$. An algorithm $\mathcal{A} : X^{\mathbb{R}} \to X^T$ that takes as input a sequence of loss functions and outputs decisions such that $\sum_{t=1}^T \frac{1}{T}\ell^{(t)}(\mathcal{A}_t(\{\ell^{(t)}\}_{t=1}^T)) - \min_{\boldsymbol{x} \in X} \sum_{t=1}^T \frac{1}{T}\ell^{(t)}(\boldsymbol{x}) \to 0$ as $T \to \infty$ is called a **no-regret algorithm**.

A first-order method that solves OCPs is **Online Mirror Descent (OMD)**. For some initial iterates $\boldsymbol{u}^{(0)} = 0$ and $\boldsymbol{x}^{(t)} \in X$, OMD performs the following update in the dual space $X^*$ at each time step $t$: $\boldsymbol{u}^{(t+1)} = \boldsymbol{u}^{(t)} - \eta \nabla_{\boldsymbol{x}} \ell^{(t)}(\boldsymbol{x}^{(t)})$, and then projects the iterate computed in the dual space $X^*$ back to the primal space $X$: $\boldsymbol{x}^{(t+1)} = \arg\min_{\boldsymbol{x} \in X} \{R(\boldsymbol{x}) - \langle \boldsymbol{u}^{(t+1)}, \boldsymbol{x} \rangle\}$, where $R : X \to \mathbb{R}$ is a strongly-convex differentiable function. When $R(\boldsymbol{x}) = \frac{1}{2}\|\boldsymbol{x}\|_2^2$, OMD reduces to **projected online gradient descent**, given by the update rule: $\boldsymbol{x}^{(t+1)} = \Pi_X\left(\boldsymbol{x}^{(t)} - \eta \nabla_{\boldsymbol{x}} \ell^{(t)}(\boldsymbol{x}^{(t)})\right)$. The following theorem bounds the **average regret** of OMD (Kakade, Shalev-Shwartz, and Tewari 2012):

**Theorem 12.** *Let $c = \max_{\boldsymbol{x} \in X} \|\boldsymbol{x}\|$, and let $\{\ell^{(t)}\}_t$ be a sequence of $\ell$-Lipschitz loss functions s.t. for all $t \in \mathbb{N}_+$, $\ell^{(t)} : \mathbb{R}^n \to \mathbb{R}$ with respect to the dual norm $\|\cdot\|_*$. Then, if $\eta = \frac{c}{\ell\sqrt{2T}}$, online projected gradient descent achieves bounded average regret bounded as follows:*
$$\sum_{t=1}^T \frac{1}{T}\ell^{(t)}(\boldsymbol{x}^{(t)}) - \min_{\boldsymbol{x} \in X} \sum_{t=1}^T \frac{1}{T}\ell^{(t)}(\boldsymbol{x}) \leq c\ell\sqrt{\frac{2}{T}}.$$

# B  Additional Related Work

**Related Work**  Stackelberg games (Von Stackelberg 1934) have found important applications in the domain of security (e.g., (Nguyen et al. 2016; Sinha et al. 2018)) and environmental protection (e.g., (Fang and Nguyen 2016)). These applications have thus far been modelled as Stackelberg games with independent strategy sets. Yet, the increased expressiveness of Stackelberg games with dependent strategy sets may make them a better model of the real world, as they provide the leader with more power to achieve a better outcome by constraining the follower's choices.

The study of algorithms that compute competitive equilibria in Fisher markets was initiated by Devanur et al. (Devanur et al. 2002), who provided a polynomial-time method for solving these markets assuming linear utilities. More recently, Cheung, Hoefer, and Nakhe (Cheung, Hoefer, and Nakhe 2019) studied two price adjustment processes, tâtonnement and proportional response dynamics, in dynamic Fisher markets and showed that these price adjustment processes track the equilibrium of Fisher markets closely even when the market is subject to change.

**Additional Related Work**  Much progress has been made recently in solving min-max games with independent strategy sets, both in the convex-concave case and in non-convex-concave case. We provide a survey of the literature as presented by Goktas and Greenwald in what follows. For the former case, when $f$ is $\mu_{\boldsymbol{x}}$-strongly-convex in $\boldsymbol{x}$ and $\mu_{\boldsymbol{y}}$-strongly-concave in $\boldsymbol{y}$, Tseng (Tseng 1995), Yurii Nesterov (Yurii Nesterov 2011), and Gidel et al. (Gidel et al. 2020) proposed variational inequality methods, and Mokhtari, Ozdaglar, and Pattathil (Mokhtari, Ozdaglar, and Pattathil 2020), gradient-descent-ascent (GDA)-based methods, all of which compute a solution in $\tilde{O}(\mu_{\boldsymbol{y}} + \mu_{\boldsymbol{x}})$ iterations. These upper bounds were recently complemented by the lower bound of $\tilde{\Omega}(\sqrt{\mu_{\boldsymbol{y}}\mu_{\boldsymbol{x}}})$, shown by Ibrahim et al. (Ibrahim et al. 2019) and Zhang, Hong, and Zhang (Zhang, Hong, and Zhang 2020). Subsequently, Lin, Jin, and Jordan (Lin, Jin, and Jordan 2020b) and Alkousa et al. (Alkousa et al. 2020) analyzed algorithms that converge in $\tilde{O}(\sqrt{\mu_{\boldsymbol{y}}\mu_{\boldsymbol{x}}})$ and $\tilde{O}(\min\{\mu_{\boldsymbol{x}}\sqrt{\mu_{\boldsymbol{y}}}, \mu_{\boldsymbol{y}}\sqrt{\mu_{\boldsymbol{x}}}\})$ iterations, respectively.

For the special case where $f$ is $\mu_{\boldsymbol{x}}$-strongly convex in $\boldsymbol{x}$ and linear in $\boldsymbol{y}$, Juditsky, Nemirovski et al. (Juditsky, Nemirovski et al. 2011), Hamedani and Aybat (Hamedani and Aybat 2018), and Zhao (Zhao 2019) all present methods that converge to an $\varepsilon$-approximate solution in $O(\sqrt{\mu_{\boldsymbol{x}}/\varepsilon})$ iterations. When the strong concavity or linearity assumptions of $f$ on $\boldsymbol{y}$ are dropped, and $f$ is assumed to be $\mu_{\boldsymbol{x}}$-strongly-convex in $\boldsymbol{x}$ but only concave in $\boldsymbol{y}$, Thekumparampil et al. (Thekumparampil et al. 2019) provide an algorithm that converges to an $\varepsilon$-approximate solution in $\tilde{O}(\mu_{\boldsymbol{x}}/\varepsilon)$ iterations, and Ouyang and Xu (Ouyang and Xu 2018) provide a lower bound of $\tilde{\Omega}\left(\sqrt{\mu_{\boldsymbol{x}}/\varepsilon}\right)$ iterations on this same computation. Lin, Jin, and Jordan then went on to develop a faster algorithm, with iteration complexity of $\tilde{O}\left(\sqrt{\mu_{\boldsymbol{x}}/\varepsilon}\right)$, under the same conditions.

When $f$ is simply assumed to be convex-concave, Nemirovski (Nemirovski 2004), Nesterov (Nesterov 2007), and Tseng (Tseng 2008) describe algorithms that solve for an $\varepsilon$-approximate solution with $\tilde{O}\left(\varepsilon^{-1}\right)$ iteration complexity, and Ouyang and Xu (Ouyang and Xu 2018) prove a corresponding lower bound of $\Omega(\varepsilon^{-1})$.

When $f$ is assumed to be non-convex-$\mu_{\boldsymbol{y}}$-strongly-concave, and the goal is to compute a first-order Nash, Sanjabi et al. (Sanjabi et al. 2018b) provide an algorithm that converges to $\varepsilon$-an approximate solution in $O(\varepsilon^{-2})$ iterations. Jin, Netrapalli, and Jordan (Jin, Netrapalli, and Jordan 2020), Rafique et al. (Rafique et al. 2019), Lin, Jin, and Jordan (Lin, Jin, and Jordan 2020a), and Lu, Tsaknakis, and Hong (Lu, Tsaknakis, and Hong 2019) provide algorithms that converge in $\tilde{O}\left(\mu_{\boldsymbol{y}}^2\varepsilon^{-2}\right)$ iterations, while Lin, Jin, and Jordan (Lin, Jin, and Jordan 2020b) provide an even faster algorithm, with an iteration complexity of $\tilde{O}\left(\sqrt{\mu_{\boldsymbol{y}}}\varepsilon^{-2}\right)$.

When $f$ is non-convex-non-concave and the goal to compute is an approximate first-order Nash equilibrium, Lu, Tsaknakis, and Hong (Lu, Tsaknakis, and Hong 2019) provide an algorithm with iteration complexity $\tilde{O}(\varepsilon^{-4})$, while Nouiehed et al. (Nouiehed et al. 2019) provide an algorithm with iteration complexity $\tilde{O}(\varepsilon^{-3.5})$. More recently, Ostrovskii, Lowy, and Razaviyayn (Ostrovskii, Lowy, and Razaviyayn 2020) and Lin, Jin, and Jordan (Lin, Jin, and Jordan 2020b) proposed an algorithm with iteration complexity $\tilde{O}\left(\varepsilon^{-2.5}\right)$.

When $f$ is non-convex-non-concave and the desired solution concept is a "local" Stackelberg equilibrium, Jin, Netrapalli, and Jordan (Jin, Netrapalli, and Jordan 2020), Rafique et al. (Rafique et al. 2019), and Lin, Jin, and Jordan (Lin, Jin, and Jordan 2020a) provide algorithms with a $\tilde{O}\left(\varepsilon^{-6}\right)$ complexity. More recently, Thekumparampil et al. (Thekumparampil et al. 2019), Zhao (Zhao 2020), and Lin, Jin, and Jordan (Lin, Jin, and Jordan 2020b) have proposed algorithms that converge to an $\varepsilon$-approximate solution in $\tilde{O}\left(\varepsilon^{-3}\right)$ iterations.

We summarize the literature pertaining to the convex-concave and the non-convex-concave settings in Tables 1 and 2 respectively.

Table 1: Iteration complexities for min-max games with independent strategy sets in convex-concave settings. Note that these results assume that the objective function is Lipschitz-smooth.

| Setting | Reference | Iteration Complexity |
|---|---|---|
| $\mu_{\boldsymbol{x}}$-Strongly-Convex-$\mu_{\boldsymbol{y}}$-Strongly-Concave | (Tseng 1995) | $\tilde{O}\left(\mu_{\boldsymbol{x}} + \mu_{\boldsymbol{y}}\right)$ |
| | (Yurii Nesterov 2011) | |
| | (Gidel et al. 2020) | |
| | (Mokhtari, Ozdaglar, and Pattathil 2020) | |
| | (Alkousa et al. 2020) | $\tilde{O}(\min\left\{\mu_{\boldsymbol{x}}\sqrt{\mu_{\boldsymbol{y}}}, \mu_{\boldsymbol{y}}\sqrt{\mu_{\boldsymbol{x}}}\right])\}$ |
| | (Lin, Jin, and Jordan 2020b) | $\tilde{O}(\sqrt{\mu_{\boldsymbol{x}}\mu_{\boldsymbol{y}}})$ |
| | (Ibrahim et al. 2019) | $\tilde{\Omega}(\sqrt{\mu_{\boldsymbol{x}}\mu_{\boldsymbol{y}}})$ |
| | (Zhang, Hong, and Zhang 2020) | |
| $\mu_{\boldsymbol{x}}$-Strongly-Convex-Linear | (Juditsky, Nemirovski et al. 2011) | $O\left(\sqrt{\mu_{\boldsymbol{x}}/\varepsilon}\right)$ |
| | (Hamedani and Aybat 2018) | |
| | (Zhao 2019) | |
| $\mu_{\boldsymbol{x}}$-Strongly-Convex-Concave | (Thekumparampil et al. 2019) | $\tilde{O}\left(\mu_{\boldsymbol{x}}/\sqrt{\varepsilon}\right)$ |
| | (Lin, Jin, and Jordan 2020b) | $\tilde{O}(\sqrt{\mu_{\boldsymbol{x}}/\varepsilon})$ |
| | (Ouyang and Xu 2018) | $\tilde{\Omega}\left(\sqrt{\mu_{\boldsymbol{x}}/\varepsilon}\right)$ |
| Convex-Concave | (Nemirovski 2004) | $O\left(\varepsilon^{-1}\right)$ |
| | (Nesterov 2007) | |
| | (Tseng 2008) | |
| | (Lin, Jin, and Jordan 2020b) | $\tilde{O}\left(\varepsilon^{-1}\right)$ |
| | (Ouyang and Xu 2018) | $\Omega(\varepsilon^{-1})$ |

Table 2: Iteration complexities for min-max games with independent strategy sets in non-convex-concave settings. Note that although all these results assume that the objective function is Lipschitz-smooth, some authors make additional assumptions: e.g., (Nouiehed et al. 2019) obtain their result for objective functions that satisfy the Lojasiwicz condition.

| Setting | Reference | Iteration Complexity |
|---|---|---|
| Nonconvex-$\mu_{\boldsymbol{y}}$-Strongly-Concave, First Order Nash or Local Stackelberg Equilibrium | (Jin, Netrapalli, and Jordan 2020) | $\tilde{O}(\mu_{\boldsymbol{y}}^2 \varepsilon^{-2})$ |
| | (Rafique et al. 2019) | |
| | (Lin, Jin, and Jordan 2020a) | |
| | (Lu, Tsaknakis, and Hong 2019) | |
| | (Lin, Jin, and Jordan 2020b) | $\tilde{O}\left(\sqrt{\mu_{\boldsymbol{y}}}\varepsilon^{-2}\right)$ |
| Nonconvex-Concave, First Order Nash Equilibrium | (Lu, Tsaknakis, and Hong 2019) | $\tilde{O}\left(\varepsilon^{-4}\right)$ |
| | (Nouiehed et al. 2019) | $\tilde{O}\left(\varepsilon^{-3.5}\right)$ |
| | (Ostrovskii, Lowy, and Razaviyayn 2020) | $\tilde{O}\left(\varepsilon^{-2.5}\right)$ |
| | (Lin, Jin, and Jordan 2020b) | |
| Nonconvex-Concave, Local Stackelberg Equilibrium | (Jin, Netrapalli, and Jordan 2020) | $\tilde{O}(\varepsilon^{-6})$ |
| | (Nouiehed et al. 2019) | |
| | (Lin, Jin, and Jordan 2020b) | |
| | (Thekumparampil et al. 2019) | $\tilde{O}(\varepsilon^{-3})$ |
| | (Zhao 2020) | |
| | (Lin, Jin, and Jordan 2020b) | |

# C Omitted Proofs

*Proof of Theorem 3.* Since pessimistic regret is bounded by $\varepsilon$ after $T$ iterations, it holds that:

$$\max_{\boldsymbol{x}\in X} \text{PesRegret}_X^{(T)}(\boldsymbol{x}) \leq \varepsilon \qquad (11)$$

$$\frac{1}{T}\sum_{t=1}^T V_X^{(t)}(\boldsymbol{x}^{(t)}) - \min_{\boldsymbol{x}\in X}\sum_{t=1}^T \frac{1}{T}V_X^{(t)}(\boldsymbol{x}) \leq \varepsilon \qquad (12)$$

Since the game is static, and it further holds that:

$$\frac{1}{T}\sum_{t=1}^T V_X(\boldsymbol{x}^{(t)}) - \min_{\boldsymbol{x}\in X}\sum_{t=1}^T \frac{1}{T}V_X(\boldsymbol{x}) \leq \varepsilon \qquad (13)$$

$$\frac{1}{T}\sum_{t=1}^T V_X(\boldsymbol{x}^{(t)}) - \min_{\boldsymbol{x}\in X} V_X(\boldsymbol{x}) \leq \varepsilon \qquad (14)$$

Thus, by the convexity of $V_X$ (see Proposition 2), $V_X(\bar{\boldsymbol{x}}^{(T)}) - \min_{\boldsymbol{x}\in X} V_X(\boldsymbol{x}) \leq \varepsilon$. Now replacing $V_X$ by its definition, and setting $\boldsymbol{y}^*(\bar{\boldsymbol{x}}^{(T)}) \in \text{BR}_Y(\bar{\boldsymbol{x}}^{(T)})$, we obtain that $\left(\bar{\boldsymbol{x}}^{(T)}, \boldsymbol{y}^*(\bar{\boldsymbol{x}}^{(T)})\right)$ is $(\varepsilon, 0)$-Stackelberg equilibrium:

$$V_X(\bar{\boldsymbol{x}}^{(T)}) \leq f(\bar{\boldsymbol{x}}^{(T)}, \boldsymbol{y}^*(\bar{\boldsymbol{x}}^{(T)}))$$
$$\leq \min_{\boldsymbol{x}\in X} V_X(\boldsymbol{x}) + \varepsilon \qquad (15)$$

$$\max_{\boldsymbol{y}\in Y: \boldsymbol{g}(\bar{\boldsymbol{x}}^{(T)}, \boldsymbol{y})} f(\bar{\boldsymbol{x}}^{(T)}, \boldsymbol{y}) \leq f(\bar{\boldsymbol{x}}^{(T)}, \boldsymbol{y}^*(\bar{\boldsymbol{x}}^{(T)}))$$
$$\leq \min_{\boldsymbol{x}\in X}\max_{\boldsymbol{y}\in Y: \boldsymbol{g}(\boldsymbol{x}, \boldsymbol{y})} f(\boldsymbol{x}, \boldsymbol{y}) + \varepsilon \qquad (16)$$

$\square$

*Proof of Lemma 5.* We can relax the inner player's payoff maximization problem via the problem's Lagrangian and since by assumption 1, Slater's condition is satisfied, strong duality holds, giving us for all $\boldsymbol{x} \in X$:
$\max_{\boldsymbol{y}\in Y: \boldsymbol{g}(\boldsymbol{x},\boldsymbol{y})\geq \boldsymbol{0}} f(\boldsymbol{x}, \boldsymbol{y}) = \max_{\boldsymbol{y}\in Y}\min_{\boldsymbol{\lambda}\geq \boldsymbol{0}} \mathcal{L}_{\boldsymbol{x}}(\boldsymbol{y}, \boldsymbol{\lambda})$
$= \min_{\boldsymbol{\lambda}\geq \boldsymbol{0}}\max_{\boldsymbol{y}\in Y} \mathcal{L}_{\boldsymbol{x}}(\boldsymbol{y}, \boldsymbol{\lambda})$. We can then re-express the min-max game as: $\min_{\boldsymbol{x}\in X}\max_{\boldsymbol{y}\in Y: \boldsymbol{g}(\boldsymbol{x},\boldsymbol{y})\geq \boldsymbol{0}} f(\boldsymbol{x}, \boldsymbol{y}) = \min_{\boldsymbol{\lambda}\geq \boldsymbol{0}}\min_{\boldsymbol{x}\in X}\max_{\boldsymbol{y}\in Y}$
$\mathcal{L}_{\boldsymbol{x}}(\boldsymbol{y}, \boldsymbol{\lambda})$. Letting $\boldsymbol{\lambda}^* \in$
$\arg\min_{\boldsymbol{\lambda}\geq \boldsymbol{0}}\min_{\boldsymbol{x}\in X}\max_{\boldsymbol{y}\in Y} \mathcal{L}_{\boldsymbol{x}}(\boldsymbol{y}, \boldsymbol{\lambda})$, we have $\min_{\boldsymbol{x}\in X}$
$\max_{\boldsymbol{y}\in Y: \boldsymbol{g}(\boldsymbol{x},\boldsymbol{y})\geq \boldsymbol{0}} f(\boldsymbol{x}, \boldsymbol{y}) = \min_{\boldsymbol{x}\in X}\max_{\boldsymbol{y}\in Y} \mathcal{L}_{\boldsymbol{x}}(\boldsymbol{y}, \boldsymbol{\lambda}^*)$.
Note that $\mathcal{L}_{\boldsymbol{x}}(\boldsymbol{y}, \boldsymbol{\lambda}^*)$ is convex-concave in $(\boldsymbol{x}, \boldsymbol{y})$. Hence, any Stackelberg equilibrium $(\boldsymbol{x}^*, \boldsymbol{y}^*) \in X \times Y$ of $(X, Y, f, \boldsymbol{g})$ is a saddle point of $\mathcal{L}_{\boldsymbol{x}}(\boldsymbol{y}, \boldsymbol{\lambda}^*)$, i.e., $\forall \boldsymbol{x} \in X, \boldsymbol{y} \in Y, \mathcal{L}_{\boldsymbol{x}^*}(\boldsymbol{y}, \boldsymbol{\lambda}^*) \leq \mathcal{L}_{\boldsymbol{x}^*}(\boldsymbol{y}^*, \boldsymbol{\lambda}^*) \leq \mathcal{L}_{\boldsymbol{x}}(\boldsymbol{y}^*, \boldsymbol{\lambda}^*)$. $\square$

*Proof of Theorem 6.* Since the Lagrangian regret is bounded

for both players we have:

$$\begin{cases} \max_{\boldsymbol{x} \in X} \text{LagrRegret}_X^{(T)}(\boldsymbol{x}) \leq \varepsilon \\ \max_{\boldsymbol{y} \in Y} \text{LagrRegret}_Y^{(T)}(\boldsymbol{y}) \leq \varepsilon \end{cases} \quad (17)$$

$$\begin{cases} \frac{1}{T} \sum_{t=1}^{T} \mathcal{L}_{x^{(t)}}^{(t)}(\boldsymbol{y}^{(t)}, \boldsymbol{\lambda}^*) - \min_{\boldsymbol{x} \in X} \frac{1}{T} \sum_{t=1}^{T} \mathcal{L}_{\boldsymbol{x}}^{(t)}(\boldsymbol{y}^{(t)}, \boldsymbol{\lambda}^*) \leq \varepsilon \\ \max_{\boldsymbol{y} \in Y} \frac{1}{T} \sum_{t=1}^{T} \mathcal{L}_{\boldsymbol{x}^{(t)}}^{(t)}(\boldsymbol{y}, \boldsymbol{\lambda}^*) - \frac{1}{T} \sum_{t=1}^{T} \mathcal{L}_{\boldsymbol{x}^{(t)}}^{(t)}(\boldsymbol{y}^{(t)}, \boldsymbol{\lambda}^*) \leq \varepsilon \end{cases}$$
$$(18)$$

$$\begin{cases} \frac{1}{T} \sum_{t=1}^{T} \mathcal{L}_{x^{(t)}}(\boldsymbol{y}^{(t)}, \boldsymbol{\lambda}^*) - \min_{\boldsymbol{x} \in X} \frac{1}{T} \sum_{t=1}^{T} \mathcal{L}_{\boldsymbol{x}}(\boldsymbol{y}^{(t)}, \boldsymbol{\lambda}^*) \leq \varepsilon \\ \max_{\boldsymbol{y} \in Y} \frac{1}{T} \sum_{t=1}^{T} \mathcal{L}_{\boldsymbol{x}^{(t)}}(\boldsymbol{y}, \boldsymbol{\lambda}^*) - \frac{1}{T} \sum_{t=1}^{T} \mathcal{L}_{\boldsymbol{x}^{(t)}}(\boldsymbol{y}^{(t)}, \boldsymbol{\lambda}^*) \leq \varepsilon \end{cases}$$
$$(19)$$

The last line follows because the min-max Stackelberg game is static.

Summing the final two inequalities yields:

$$\max_{\boldsymbol{y} \in Y} \frac{1}{T} \sum_{t=1}^{T} \mathcal{L}_{\boldsymbol{x}^{(t)}}(\boldsymbol{y}, \boldsymbol{\lambda}^*) - \min_{\boldsymbol{x} \in X} \frac{1}{T} \sum_{t=1}^{T} \mathcal{L}_{\boldsymbol{x}}(\boldsymbol{y}^{(t)}, \boldsymbol{\lambda}^*) \leq 2\varepsilon$$
$$(20)$$

$$\frac{1}{T} \sum_{t=1}^{T} \max_{\boldsymbol{y} \in Y} \mathcal{L}_{\boldsymbol{x}^{(t)}}(\boldsymbol{y}, \boldsymbol{\lambda}^*) - \frac{1}{T} \sum_{t=1}^{T} \min_{\boldsymbol{x} \in X} \mathcal{L}_{\boldsymbol{x}}(\boldsymbol{y}^{(t)}, \boldsymbol{\lambda}^*) \leq 2\varepsilon$$
$$(21)$$

where the second inequality was obtained by an application of Jensen's inequality on the first and second terms.

Since $\mathcal{L}$ is convex in $\boldsymbol{x}$ and concave in $\boldsymbol{y}$, we have that $\max_{\boldsymbol{y} \in Y} \mathcal{L}_{\boldsymbol{x}^{(t)}}(\boldsymbol{y}, \boldsymbol{\lambda}^*)$ is convex in $\boldsymbol{x}$ and $\min_{\boldsymbol{x} \in X} \mathcal{L}_{\boldsymbol{x}}(\boldsymbol{y}^{(t)}, \boldsymbol{\lambda}^*)$ is convex in $\boldsymbol{y}$, which implies that $\max_{\boldsymbol{y} \in Y} \mathcal{L}_{\bar{\boldsymbol{x}}^{(T)}}(\boldsymbol{y}, \boldsymbol{\lambda}^*) - \min_{\boldsymbol{x} \in X} \mathcal{L}_{\boldsymbol{x}}(\bar{\boldsymbol{y}}^{(T)}, \boldsymbol{\lambda}^*) \leq 2\varepsilon$. By the max-min inequality ((Boyd, Boyd, and Vandenberghe 2004), Equation 5.46), it also holds that $\min_{\boldsymbol{x} \in X} \mathcal{L}_{\boldsymbol{x}}(\bar{\boldsymbol{y}}^{(T)}, \boldsymbol{\lambda}^*) \leq \max_{\boldsymbol{y} \in Y} \mathcal{L}_{\bar{\boldsymbol{x}}^{(T)}}(\boldsymbol{y}, \boldsymbol{\lambda}^*)$. Combining these two inequality yields the desired result. $\square$

*Proof of Theorem 10.* The value function of the outer player in the game $\{(X, Y, f^{(t)})\}_{t=1}^{T}$ at iteration $t \in [T]$, is given by $V^{(t)}(\boldsymbol{x}) = \max_{\boldsymbol{y} \in Y} f^{(t)}(\boldsymbol{x}, \boldsymbol{y})$. Hence, for all $t \in [T]$, as $f^{(t)}$ is $\mu$-strongly-convex, $V^{(t)}$ is also strongly concave since the maximum preserves strong-convexity.

Additionally, since for all $t \in [T]$, $f^{(t)}$ is strictly concave in $\boldsymbol{y}$, by Danskin's theorem (Danskin 1966), for all $t \in [T]$, $V^{(t)}$ is differentiable and its derivative is given by $\nabla_{\boldsymbol{x}} V^{(t)}(\boldsymbol{x}) = \nabla_{\boldsymbol{x}} f(\boldsymbol{x}, \boldsymbol{y}^*(\boldsymbol{x}))$ where $\boldsymbol{y}^*(\boldsymbol{x}) \in \max_{\boldsymbol{y} \in Y} f^{(t)}(\boldsymbol{x}, \boldsymbol{y})$. Thus, as $\nabla_{\boldsymbol{x}} f(\boldsymbol{x}, \boldsymbol{y}^*(\boldsymbol{x}))$ is $\ell_{\nabla f}$-lipschitz continuous, so is $\nabla_{\boldsymbol{x}} V^{(t)}(\boldsymbol{x})$. The result follows from Cheung, Hoefer, and Nakhe's bound for gradient descent on shifting strongly convex functions ((Cheung, Hoefer, and Nakhe 2019), Proposition 12).
$\square$

*Proof of Theorem 11.* By the assumptions of the theorem, the loss functions of the outer player $\{f^{(t)}(\cdot, \boldsymbol{y}^{(t)})\}_{t=1}^{T}$ are $\mu_{\boldsymbol{x}}$-strongly-convex and $\ell_{\nabla f}$-Lipschitz continuous

functions. Similarly the loss functions of the inner player $\{-f^{(t)}(\boldsymbol{x}^{(t)}, \cdot)\}_{t=1}^{T}$ are $\mu_{\boldsymbol{y}}$-strongly-convex and $\ell_{\nabla f}$-Lipschitz continuous functions. Using Cheung, Hoefer, and Nakhe's Proposition 12 (Cheung, Hoefer, and Nakhe 2019), we then obtain the following bounds:

$$\left\| \boldsymbol{x}^{(T)^*} - \boldsymbol{x}^{(T)} \right\| \leq (1 - \delta_{\boldsymbol{x}})^{T/2} \left\| \boldsymbol{x}^{(0)^*} - \boldsymbol{x}^{(0)} \right\|$$
$$+ \sum_{t=1}^{T} (1 - \delta_{\boldsymbol{x}})^{\frac{T-t}{2}} \Delta_{\boldsymbol{x}}^{(t)} \quad (22)$$

$$\left\| \boldsymbol{y}^{(T)^*} - \boldsymbol{y}^{(T)} \right\| \leq (1 - \delta_{\boldsymbol{y}})^{T/2} \left\| \boldsymbol{y}^{(0)^*} - \boldsymbol{y}^{(0)} \right\|$$
$$+ \sum_{t=1}^{T} (1 - \delta_{\boldsymbol{y}})^{\frac{T-t}{2}} \Delta_{\boldsymbol{y}}^{(t)} \quad (23)$$

Combining the two inequalities, we obtain:

$$\left\| \boldsymbol{x}^{(T)^*} - \boldsymbol{x}^{(T)} \right\| + \left\| \boldsymbol{y}^{(T)^*} - \boldsymbol{y}^{(T)} \right\|$$
$$\leq (1 - \delta_{\boldsymbol{x}})^{T/2} \left\| \boldsymbol{x}^{(0)^*} - \boldsymbol{x}^{(0)} \right\| + (1 - \delta_{\boldsymbol{y}})^{T/2} \left\| \boldsymbol{y}^{(0)^*} - \boldsymbol{y}^{(0)} \right\|$$
$$+ \sum_{t=1}^{T} (1 - \delta_{\boldsymbol{x}})^{\frac{T-t}{2}} \Delta_{\boldsymbol{x}}^{(t)} + \sum_{t=1}^{T} (1 - \delta_{\boldsymbol{y}})^{\frac{T-t}{2}} \Delta_{\boldsymbol{y}}^{(t)} \quad (24)$$

The second part of the theorem follows by taking the sum of the geometric series. $\square$

# D  Pseudo-Code for Algorithms

---

**Algorithm 1: Max-Oracle Gradient Descent**

---

**Inputs:** $X, Y, f, \boldsymbol{g}, \boldsymbol{\eta}, T, \boldsymbol{x}^{(0)}$
**Output:** $(\boldsymbol{x}^*, \boldsymbol{y}^*)$

1: **for** $t = 1, \ldots, T$ **do**
2:     Find $\boldsymbol{y}^*(\boldsymbol{x}^{(t-1)}) \in \mathrm{BR}_Y(\boldsymbol{x}^{(t-1)})$
3:     Set $\boldsymbol{y}^{(t-1)} = \boldsymbol{y}^*(\boldsymbol{x}^{(t-1)})$
4:     Set $\boldsymbol{\lambda}^{(t-1)} = \boldsymbol{\lambda}^*(\boldsymbol{x}^{(t-1)}, \boldsymbol{y}^{(t-1)})$
5:     Set $\boldsymbol{x}^{(t)} = \Pi_X\left[\boldsymbol{x}^{(t-1)} - \eta_t \nabla_{\boldsymbol{x}} \mathcal{L}_{\boldsymbol{x}^{(t-1)}}\left(\boldsymbol{y}^{(t-1)}, \boldsymbol{\lambda}^{(t-1)}\right)\right]$
6: **end for**
7: Set $\bar{\boldsymbol{x}}^{(T)} = \frac{1}{T}\sum_{t=1}^{T} \boldsymbol{x}^{(t)}$
8: Set $\boldsymbol{y}^*(\bar{\boldsymbol{x}}^{(T)}) \in \mathrm{BR}_Y(\bar{\boldsymbol{x}}^{(T)})$
9: **return** $(\bar{\boldsymbol{x}}^{(T)}, \boldsymbol{y}^*(\bar{\boldsymbol{x}}^{(T)}))$

---

**Algorithm 2: Lagrangian Gradient Descent Ascent (LGDA)**

---

**Inputs:** $\boldsymbol{\lambda}^*, X, Y, f, \boldsymbol{g}, \boldsymbol{\eta^x}, \boldsymbol{\eta^y}, T, \boldsymbol{x}^{(0)}, \boldsymbol{y}^{(0)}$
**Output:** $\boldsymbol{x}^*, \boldsymbol{y}^*$

1: **for** $t = 1, \ldots, T-1$ **do**
2:     Set $\boldsymbol{x}^{(t+1)} = \Pi_X\left(\boldsymbol{x}^{(t)} - \eta_t^{\boldsymbol{x}} \nabla_{\boldsymbol{x}} \mathcal{L}_{\boldsymbol{x}^{(t)}}(\boldsymbol{y}^{(t)}, \boldsymbol{\lambda}^*)\right)$
3:     Set $\boldsymbol{y}^{(t+1)} = \Pi_Y\left(\boldsymbol{y}^{(t)} + \eta_t^{\boldsymbol{y}} \nabla_{\boldsymbol{y}} \mathcal{L}_{\boldsymbol{x}^{(t)}}(\boldsymbol{y}^{(t)}, \boldsymbol{\lambda}^*)\right)$
4: **end for**
5: **return** $\{(\boldsymbol{x}^{(t)}, \boldsymbol{y}^{(t)})\}_{t=1}^{T}$

---

**Algorithm 3: Dynamic tâtonnement**

---

**Inputs:** $T, \{(U^{(t)}, \boldsymbol{b}^{(t)}, \boldsymbol{s}^{(t)})\}_{t=1}^{T}, \boldsymbol{\eta}, \boldsymbol{p}^{(0)}, \delta$
**Output:** $\boldsymbol{x}^\star, \boldsymbol{y}^\star$

1: **for** $t = 1, \ldots, T-1$ **do**
2:     For all $i \in [n]$, find $\boldsymbol{x}_i^{(t)} \in \arg\max_{\boldsymbol{x}_i \in \mathbb{R}_+^m : \boldsymbol{x}_i \cdot \boldsymbol{p}^{(t-1)} \le b_i^{(t)}} u_i(\boldsymbol{x}_i)$
3:     Set $\boldsymbol{p}^{(t)} = \Pi_{\mathbb{R}_+^m}\left(\boldsymbol{p}^{(t-1)} - \eta_t(\boldsymbol{s}^{(t)} - \sum_{i \in [n]} \boldsymbol{x}_i^{(t)})\right)$
4: **end for**
5: **return** $(\boldsymbol{p}^{(t)}, \boldsymbol{X}^{(t)})_{t=1}^{T}$

---

**Algorithm 4: Dynamic Myopic Best-Response Dynamics**

---

**Inputs:** $\{(U^{(t)}, \boldsymbol{b}^{(t)}, \boldsymbol{s}^{(t)})\}_{t=1}^{T}, \boldsymbol{\eta^p}, \boldsymbol{\eta^X}, T, \boldsymbol{X}^{(0)}, \boldsymbol{p}^{(0)}$
**Output:** $\boldsymbol{x}^\star, \boldsymbol{y}^\star$

1: **for** $t = 1, \ldots, T-1$ **do**
2:     Set $\boldsymbol{p}^{(t+1)} = \Pi_{\mathbb{R}_+^m}\left(\boldsymbol{p}^{(t)} - \eta_t^{\boldsymbol{p}}(\boldsymbol{s}^{(t)} - \sum_{i \in [n]} \boldsymbol{x}_i^{(t)})\right)$
3:     For all $i \in [n]$, set $\boldsymbol{x}_i^{(t+1)} = \Pi_{\mathbb{R}_+^m}\left(\boldsymbol{x}_i^{(t)} + \eta_t^{\boldsymbol{X}}\left(\frac{b_i^{(t)}}{u_i^{(t)}\left(\boldsymbol{x}_i^{(t)}\right)} \nabla_{\boldsymbol{x}_i} u_i^{(t)}\left(\boldsymbol{x}_i^{(t)}\right) - \boldsymbol{p}^{(t)}\right)\right)$
4: **end for**
5: **return** $(\boldsymbol{p}^{(t)}, \boldsymbol{X}^{(t)})_{t=1}^{T}$

---

# E  An Economic Application: Details

Our experimental goal was to understand if Algorithm 3 and Algorithm 4 converges in terms of distance to equilibrium and if so how the rate of convergences changes under different utility structures, i.e. different smoothness and convexity properties of the value functions.

To answer these questions, we ran multiple experiments, each time recording the prices and allocations computed by Algorithm 3, in pessimistic learning setting, and by Algorithm 4, in optimistic learning setting, during each iteration $t$ of the loop. Moreover, at each iteration $t$, we solve the competitive equilibrium $(\boldsymbol{p}^{(t)^\star}, \boldsymbol{X}^{(t)^\star})$ for the Fisher market $(U^{(t)}, \boldsymbol{b}^{(t)}, \boldsymbol{s}^{(t)})$. Finally, for each run of the algorithm on each market, we then computed distance between the computed prices, allocations and the equilibrium prices, allocations, which we plot in Figure 1 and Figure 2.

**Hyperparameters** We set up 100 different linear, Cobb-Douglas, Leontief dynamic Fisher markets with random changing market parameters across time, each with 5 buyers and 8 goods, and we randomly pick one of these experiments to graph.

In our execution of Algorithm 3, buyer $i$'s budget at iteration $t$, $b_i^{(t)}$, was drawn randomly from a uniform distribution ranging from 10 to 20 (i.e., $U[10, 20]$), each buyer $i$'s valuation for good $j$ at iteration $t$, $v_{ij}^{(t)}$, was drawn randomly from $U[5, 15]$, while each good $j$'s supply at iteration $t$, $s_j^{(t)}$, was drawn randomly from $U[100, 110]$. In our execution of Algorithm 4, buyer $i$'s budget at iteration $t$, $b_i^{(t)}$, was drawn randomly from a uniform distribution ranging from 10 to 15 (i.e., $U[10, 15]$), each buyer $i$'s valuation for good $j$ at iteration $t$, $v_{ij}^{(t)}$, was drawn randomly from $U[10, 20]$, while each good $j$'s supply at iteration $t$, $s_j^{(t)}$, was drawn randomly from $U[10, 15]$.

We ran both Algorithm 3 and Algorithm 4 for 1000 iterations on linear, Cobb-Douglas, and Leontief Fisher markets. We started the algorithm with initial prices drawn randomly from $U[5, 55]$. Our theoretical results assume fixed learning rates, but since those results apply to static games while our experiments apply to dynamic Fisher markets, we selected variable learning rates. After manual hyper-parameter tuning, for Algorithm 3, we chose a dynamic learning rate of $\eta_t = \frac{1}{\sqrt{t}}$, while for Algorithm 4, we chose learning rates of $\eta_t^{\boldsymbol{x}} = \frac{5}{\sqrt{t}}$ and $\eta_t^{\boldsymbol{y}} = \frac{0.01}{\sqrt{t}}$, for all $t \in [T]$. For these choices of learning rates, we obtain empirical convergence rates close to what the theory predicts.

**Programming Languages, Packages, and Licensing** We ran our experiments in Python 3.7 (Van Rossum and Drake Jr 1995), using NumPy (Harris et al. 2020), Pandas (pandas development team 2020), and CVXPY (Diamond and Boyd 2016). Figure 1 and Figure 2 were graphed using Matplotlib (Hunter 2007).

Python software and documentation are licensed under the PSF License Agreement. Numpy is distributed under a liberal BSD license. Pandas is distributed under a new BSD license. Matplotlib only uses BSD compatible code, and its

license is based on the PSF license. CVXPY is licensed under an APACHE license.

**Implementation Details** In order to project each allocation computed onto the budget set of the consumers, i.e., $\{\boldsymbol{X} \in \mathbb{R}_+^{n \times m} \mid \boldsymbol{X} \boldsymbol{p} \leq \boldsymbol{b}\}$, we used the alternating projection algorithm for convex sets, and alternatively projected onto the sets $\mathbb{R}_+^{n \times m}$ and $\{\boldsymbol{X} \in \mathbb{R}^{n \times m} \mid \boldsymbol{X} \boldsymbol{p} \leq \boldsymbol{b}\}$.

To compute the best-response for the inner play in Algorithm 3, we used the ECOS solver, a CVXPY's first-order convex-program solvers, but if ever a runtime exception occurred, we ran the SCS solver.

When computing the distance from the demands $\boldsymbol{X}^{(t)}$ computed by our algorithms to the equilibrium demands $\boldsymbol{X}^{(t)^\star}$, we normalize both demands to satisfy $\forall j \in [m]$, $\sum_{i \in [n]} x_{ij} = 1_m$ to reduce the noise caused by changing supplies.

**Computational Resources** Our experiments were run on MacOS machine with 8GB RAM and an Apple M1 chip, and took about 2 hours to run. Only CPU resources were used.

**Code Repository** The data our experiments generated, as well as the code used to produce our visualizations, can be found in our code repository (https://anonymous.4open.science/r/Dynamic-Minmax-Games-8153/).