# OpenReview forum: "Robust No-Regret Learning in Min-Max Stackelberg Games"
_AAAI.org/2022/Workshop/AdvML — AAAI-22 AdvML Workshop LongPaper_

### Official Review · Reviewer_21Dc · 2021-11-24
**A novel work for Min-Max Stackelberg Games**

**Rating:** 6
**Confidence:** 2

**Review:**

This work considers min-max Stackelberg games. For two special settings of this problem, this paper proposes no-regret algorithms with convergence guarantee. Moreover, this work provides the theoretical analysis as well as expereimetal results of the algorithms' robustness.

However, the connection between this work and adversarial mechine learning is relatively weak. Moreover, it's better to adjust the format of some equations, like the equation in the bottom of page 2 and in the top of page 3, the Lipschitz-continuous condition in the final of sec 2,  “vanilla” regret in page 4, the objective in Example 4.

---

### Official Review · Reviewer_nG3n · 2021-11-29
**Convergence Analysis of No-Regret Learning Algorithms in Min-Max Stackelberg Games**

**Rating:** 6
**Confidence:** 3

**Review:**

This paper provides a convergence proof of no-regret learning algorithms in min-max Stackelberg games. Under certain assumptions, the authors prove that the no-regret learning algorithms will converge to a equilibrium after $T$ iterations in pessimistic and optimistic settings (with Lagrangian regret). The author then apply above theorems to OMD and derive $O(\frac{1}{\epsilon^2})$ convergence rate for those algorithms. Finally, the authors study the dynamic Stackelberg games and give theoretical proof for independent strategy sets.
One interesting question is: can these analyses be applied to two-player zero-sum games modeled by Markov Decision Process, which might be a more practical and challenging question to be considered.

Also some drawbacks must be addressed in terms of writing. Some inline functions can be adjusted for better reading, and grammar mistakes should be corrected before submission.

the average of the players’ strategies converge to a Stackelberg equilibrium. -> converges to

in average iterates. -> in average iterations.

We provide a review of related work in Appendix BThis paper is organized as follows. -> Appendix B. This

---

### Decision · Program_Chairs · 2021-12-01

**Decision:**

Accept (Long Paper)

**Comment:**

Both reviewers give positive ratings on this paper. Thus it is accepted as a long paper. Please address the reviewers' comments in the camera-ready version.